# Optimal planning and allocation of Plug-in Hybrid Electric Vehicles charging stations using a novel hybrid optimization technique

**Ayyappan Subramaniam**[1]*, **Lal Raja Singh Ravi Singh**[2]

**1** Department of Electrical and Electronics Engineering, Jai Shriram Engineering College, Tirupur, Tamil Nadu, India, **2** Department of Electrical and Electronics Engineering, KIT–Kalaignar karunanidhi Institute of Technology, Coimbatore, Tamil Nadu, India

* ayyappanme86@gmail.com

**Data Availability Statement:** All relevant data are within the manuscript and its Supporting Information files.

## Abstract

India's expanding population has necessitated the development of alternate transportation methods with electric vehicles (EVs) being the most indigenous and need for the current scenario. The major hindrance is the undue influence on the power distribution system caused by incorrect charging station setup. Renewable Energy Sources (RES) have a lower environmental impact than the non-renewable sources of energy and due to which Plug-in Hybrid Electric Vehicles (PHEV) charging stations are installed in the highest-ranking buses to facilitate their effective placements. Based on meta-heuristic optimization, this study offers an effective PHEV charging stations allocation approach for RES applications. The primary objective of the developed system is to create a charging network at a reasonable cost while maintaining the operational features of the distribution network. These trouble-sare handled by applying meta-heuristic algorithms and optimum planning based on renewable energy systems to satisfy the outcomes of the variables. As a result, by adding charging station parameters, this research proposes to conceptualize the distribution of optimal charging stationsas multiple-objectives of the problem. Furthermore, the PHEV RES and charging station location problem is handled in this study by deploying a novel hybrid algorithm termed as Atom Search Woven Aquila Optimization Algorithm (AT-AQ) that includes the ideas of both Aquila Optimizer (AO) and Atom Search Optimization (ASO) Algorithms. In reality, Aquila Optimizer is a unique population-based optimization approach energized by Aquila's behaviour when seeking prey and it solves the problems of slow convergence and local optimum trapping. According to the findings of the experiments, the proposed model outperformed the other methods in terms of minimized cost function.

## 1. Introduction

Renewable Energy Sources have gained attention presently as potential replacements for fossil fuels. This resource might be relocated closer to the load, reducing expenditures, losses, and voltage variations [1]. In the power and transportation sectors, factors such as global warming,

**Funding:** The author(s) received no specific funding for this work.

**Competing interests:** The authors have declared that no competing interests exist.

the depletion of fossil fuels, and increasing prices have led to significant shifts. Around one-fifth of global energy consumption is consumed by the mobility sector [2]. As the establishment of this resource was random, its extensive penetration into the grid might pose problems. As a result, higher-capacity energy storage technologies are required to sustain the network. In addition to solar and wind power, additional sustainable energy sources including geothermal, biogas, biomass, and low-impact hydropower are considered. Electric car batteries may be used to charge the campus or house via two-way charging, decreasing the need for non-renewable electricity from the grid. Because of their efficiency and ecologically friendly nature, plug-in PHEVs have increased its popularity [3]. PHEVs, like HEVs, are hybrids with bigger batteries. However, PHEVs can also be recharged by plugging them in, while HEVs only have built-in batteries that can be recharged while driving [4, 5].

In [6, 7], cohesive dispatch uses a DC power flow model that takes into account both transmission and distribution networks while taking into account voltage limits. The objective of levying parking lots is to diminish distribution network losses. To do this, a PHEV charging profile (PCP) for charging and discharging PHEVs is required. When designing PCP, the flattening of the household load profile is to be considered and PHEV features must be defined to attain PCP. Vehicle attributes are collected using the National Household Travel Survey (NHTS). PHEVs are calculated using both the number of EVs and PHEV penetration rate. Charging an enormous number of EVs at once creates a strain on the distribution system and, in rare circumstances, can lead to grid instability. The most successful ways to improve smart grid dependability are to properly allocate protection equipment [6, 8], DG units [9, 10], and EV/PHEV charging stations [11, 12].

The Vehicle-to-Grid (V2G) system is an EV feature that enables us to store, utilize, and distribute electrical energy. Various manufacturers are still exploring and improving these cars in terms of green technology and environmentally friendly conditions [13–15]. These EVs help not just with mobility, but also with lowering dependency on fossil fuels during peak load needs in the market's early phases [16, 17]. The simplest way to overcome this problem is to use charging synchronization. The idea is that EVs should submit data, such as battery SoC so that a system may prioritize charging demands and decide which EV should charge during this time slot, while delaying other demands to future time slots [18–21].

Over the past few years, works have been carried out in locating the charging stations for ease of charging the electric vehicles and as well allocating these charging stations for optimal charging of the energy to these PHEVs. Nojood O. Aljehane *et al.* [22] demonstrated an efficient BWO-based allocation of RES and CS for PHEVs, with the MPC-based recommended model. For comparing the performance of PHEVs with the present system, RES and CS are designated by the Black Widow Optimization (BWO) algorithm, while the Deep Stacked auto-encoder (DSAE) method determines the near-future velocity of HEVs. A large number of simulations were run, and the outcomes were analysed using various metrics. Finally, the experimental results demonstrated that the provided model outperformed the other strategies under consideration.

Mohammad Saadatmandi *et al.* [23] created a charge management approach to promote penetration of renewable resources. Their findings reveal that this program is designed to encourage consumers to use RES while minimizing the amount of power obtained from conventional generators. Hasan Mehrjerdi *et al.* [24] created the best charging amenity and scope for an EV charging station, as well as the charger nominal powers. Wind energy and energy storage devices provide the power for the charging station that is connected to the electric grid. The charging stations dimensions and functionalities have been optimized, and line reinforcement strengthens the electrical grid. To manage uncertainty in wind power, stochastic programming is used. It is a mixed-integer linear programming approach using the General

Algebraic Modeling System (GAMS) toolkit. In their study, the fastest charging speed with 116 kW, the intermediate charging speed with 84 kW, and the slow charging speed with 52 kW were found to be optimal.

Ali-Mohammad Hariri *et al*. [25] established a solitary optimization dilemma to boost smart grid sustainability by assigning distributed generation units, EV/PHEV CSs, and defending components optically at the same time, while taking three optimization variables into account as key contributions. In addition, Hierarchical optimization method (HOM) was created to investigate the three phases and seven cases that are developed based on the optimization factors, with findings produced in the Distributed Generation Allocation (DGA)-CSA-Protective Device Allocation (PDA) scheme and test results were emphasized. Due to the segmentation concept being applied, according to the test results, the suggested method can be flexible in determining which DG units and CSs are needed in terms of power flow states as well as consistency aspects.

Rouyi Chen *et al*. [26] framed the PHEV charging coordination crisis as a two-stage constrained optimization dilemma and devised an optimal charge control technique to solve it in two stages. The suggested scheme has the advantages, such as it provides the lowest total charging cost for all PHEVs while flattening the power demand curves for the grid, and is also simple to execute in practice. Numerical simulations were performed to demonstrate the efficiency of their strategy.

Mostafa Rezaeimozafar *et al*. [27] described a new method for determining the best placement and scope for RES and EV charging stations by taking into account the changes caused by EVs. An enhanced Genetic Algorithm (GA)—Particle Swarm Optimization (PSO) is developed for resolving the specified optimization trouble, and its effectiveness is evaluated compared to the Differential Evolution (DE) method for evaluating the developed approach. The research also shows that using EVs as a vigorous power source with RES can lower losses, voltage variations, the response to input constraints, and the costs incurred by administrators and recipients.

N. Tutkun *et al*. [28] employed off-grid sources to share power for charging PHEVs in order to minimize system damage when a high number of vehicles were exploited, as well as focusing on the layout of a grid-linked 30 kW photovoltaic powered PHEV charging station with configurable battery storage units. It is accomplished by using Demand Side Management (DSM) tactics to optimize PHEV charging times, which results in a highly anticipated seasonal influence on maneuver costs in solar PV-powered systems, which can be decreased to a reasonable level using well-designed optimization algorithms.

Zhaohao Ding *et al*. [29] suggested a stochastic resource planning strategy for PHEV CSs in order to optimize energy utilisation on both the demand and supply areas. On the supply side, acquisition verdicts for forward and spot markets are coordinated with internal generation resource management decisions, and demand-side scheduling makes optimal use of two types of charging loads. The effects on PHEV charging station operation turnover are proven using numerical simulation results.

Table 1. presents the survey on the optical allocation of charging station and renewable energy sources. At first, BWO Algorithm [22] assigns RES and charging sites for HEVs based on a simulation of black widow spiders, promoting mating behaviour, however, the role of machine learning tactics can be inspected to enrich future requirement precision and lower the cost of imbalance. Furthermore, it can be used in a real-time setting. Binary conventional generation (BCG) [23] flourishes a charging management programme to enhance penetration of renewable resources, but the penetration can be improved by using a charge management scheme. V2G technology [24] is quite sensitive to discharge depth, which limits the charging-discharging regime's time intervals limits that increases planning costs. The Hierarchical

**Table 1. Reviews on the optical allocation of RES and CS for PHEVs.**

| Author | Adopted methodology | Features | Challenges |
|---|---|---|---|
| Nojood O. Aljehane et al. [22] | BWO Algorithm | It assigns RES and charging sites for HEVs based on a simulation of black widow spiders promoting mating behaviour. | The role of machine learning strategies can be examined to improve future appeal attention and lower the cost of imbalance. Furthermore, it can be used in a real-time setting. |
| Mohammad Saadatmandi et al. [23] | BCG | Develops a charging management programme to increase penetration of RES. | To improve the penetration of renewable energy, a charge management scheme must be developed. |
| Hasan Mehrjerdi et al. [24] | V2G technology | It is quite sensitive to discharge depth. The V2G charging-discharging operation is optimized. | Limiting the charging-discharging regime's time intervals limits planning flexibility and increases planning costs. |
| Ali-Mohammad Hariri et al. [25] | HOM approach | Analytical EV/PHEV charging station reliability modelling is introduced. | Under varied scenarios, it is vital to establish the optimal value of the third escalation phase, which must be smaller than any derived objective of second phase. |
| Rouyi Chen et al. [26] | DSM and LVT controller | It minimizes the total charging cost for all PHEVs while flattening the grid's power demand curves. It is simple to put into practice. | A high percentage of PHEVs on the road could overburden the electrical grid during peak hours, raising PHEV charging costs. |
| Mostafa Rezaeimozafar et al. [27] | DE algorithm | It can lower losses, voltage variations, system operators' and subscribers' costs, as well as the uncertainty of input parameters. | Computationally complex. |
| N. Tutkun et al. [28] | Buck-boost converter topologies | When surplus energy develops, integrating batteries into a PV system is a costly investment that might be termed cheap energy storage. | The unit cost of classical power generators makes attaining all of the criteria challenging. |
| Zhaohao Ding et al. [29] | Stochastic Resource-Planning Scheme | The suggested scheme's volatility risk can be adequately handled. | PHEV charging stations must strike a balance between cost and risk. |

Following are the major contributions of this research study

➤ Proposes an enhanced Charging Station (CS) placement model for PHEVs.

➤ Proposes a new hybrid algorithm termed, "Atom Search Woven Aquila Optimization Algorithm (AT-AQ)".

➤ The analysis of the suggested AT-AQ is done concerning total cost or voltage parameters, and the outcomes thus obtained are compared with the traditional techniques to prove their superiority.

Optimization Method (HOM) [25] facilitated an analytical EV/PHEV charging station reliability modelling, which is crucial in determining the best value for the third optimization phase. DSM and Low Voltage Transformer (LVT) controller [26] minimizes the total charging cost for all PHEVs while flattening the gird's power demand curves and are simple to put into practice, but a high percentage of PHEVs on the road could overburden the electrical grid during peak hours, raising PHEV charging costs.

Differential Evolution algorithm [27] can lower losses, voltage variations, system operators' and subscribers' costs, as well as the uncertainty of input parameters and it can be enhanced by using high-speed wind. Buck-boost converter topologies [28] develop surplus energy when integrating batteries into a PV system, which is a costly investment that might be termed cheap energy storage, however, with traditional power generators; it is extremely complicated to meet all the criteria due to the high unit cost. Stochastic Resource-Planning Scheme [29] suggested that volatility risk can be adequately handled, however, PHEV charging stations must strike a balance between cost and risk.

For fast charging of the lithium-ion battery, a model with reinforcement learning employing deep deterministic policy gradient has also been developed [30]. Soft actor-critic DRL algorithm has also been developed in the work of Wu et al. for constrained energy management of Sauer battery [31]. Mohanty & Perli provided a Battle Royal Optimization algorithm based on the fuzzy multi-objective functions used for two-stage and simultaneous optimal allocations of electric vehicle charging stations [32]. Majhi et al. proposed a mixed-integer optimization

model to achieve a cost-effective solution for the optimal placement of dynamic charging facilities on the large road network while maintaining an acceptable state-of-charge level [33]. Xu & Huang (2022) developed a new hybrid clustering algorithm and a vehicle-pile resource assignment model that considers user preferences and requirements in the upper layer, and operational cost reduction in the lower layer [34].

Balu & Mukherjee modelled a novel strategy for obtaining the best location of EVCS/ EVBSSs in the radial distribution system. Also, the EV charger has been modelled as constant current load and the influence of EVCS/EVBSSs demand on the voltage profile, real power loss, total voltage deviation, energy loss cost [35].Wei et al. proposed to optimize the battery energy consumption and to reduce the tire slip loss simultaneously for EV charging [36].Yi et al. performed a novel data-driven approach to optimize electric vehicle (EV) public charging and translated the study area into a directed graph by partitioning it into discrete grids [37]. Cao et al. (2021) developed a dynamic programming description that could solve the optimal power-flow in respect of EV charging station demand allocation [38].

Thangaraj et al. modelled a hybrid technique to Electric Vehicle (EV) based Grid connected with Distributed Generation (DG). The proposed hybrid technique is the joint implementation of artificial longicorn transgender algorithm (ATLA) and Water strider algorithm (WSA) [39]. Chen et al. (2023) proposed a real-time hierarchical effective and efficient co-optimization control strategy for automated and connected PHEV to co-optimize vehicle velocity and energy management in urban driving scenarios [40]. Kathiravan & Rajnarayanan considered network loss minimization by the optimum placement of EVCS along with Distributed Generation [41].

An et al. analysed electric vehicle charging behavior characteristics, and investigated the EV charging problem at the scheduling level. A mathematical model for coordinated charging of EVs was proposed to minimize the total charging time for a given number of vehicles [42]. Liu et al. proposed an imitation reinforcement learning-based algorithm with optimal guidance for energy control of hybrid vehicles to accelerate the solving process [43]. Wang et al. (2023) designed a multi-agent reinforcement learning (MARL) based optimal energy-saving strategy for HEV, achieving a cooperative control on the powertrain [44]. Ahmad et al. modelled an approach to optimally place the solar-powered charging stations in a distribution network with improved voltage profile, minimum power loss and reduced cost [45].

The paper is segmented to be—Section 1 describes a brief preface to the paper, and the reviews of conventional strategies are inclined in section 2. Section 3 deliberates the system model of the proposed CS allocation system and section 4 demonstrates the phases of operation of the proposed AT-AQframework for the allocation of CS. Section 5 explains the outcomes of the proposed model and section 6 concludes the findings of the paper.

## 2. System model of the proposed charging station allocation system

PHEV charging stations installed in bus stations with the highest rating result in effective utilization of the system. Based on meta-heuristic optimization, this research study proposes a new PHEV charging station renewable energy source distribution approach. The primary objective function is to create a charging infrastructure at a reasonable cost, while sustaining the operational attributes of the distribution network. As a result, the proposed approach aims to define the RES and charging station allocation issue as a multiple-objective approach by integrating charging station characteristics. Furthermore, a new hybrid algorithm that integrates the principles of both the AO and ASO Algorithms have been developed to tackle the PHEV RES and charging station location difficulty. Aquila Optimization is rather a unique population-based

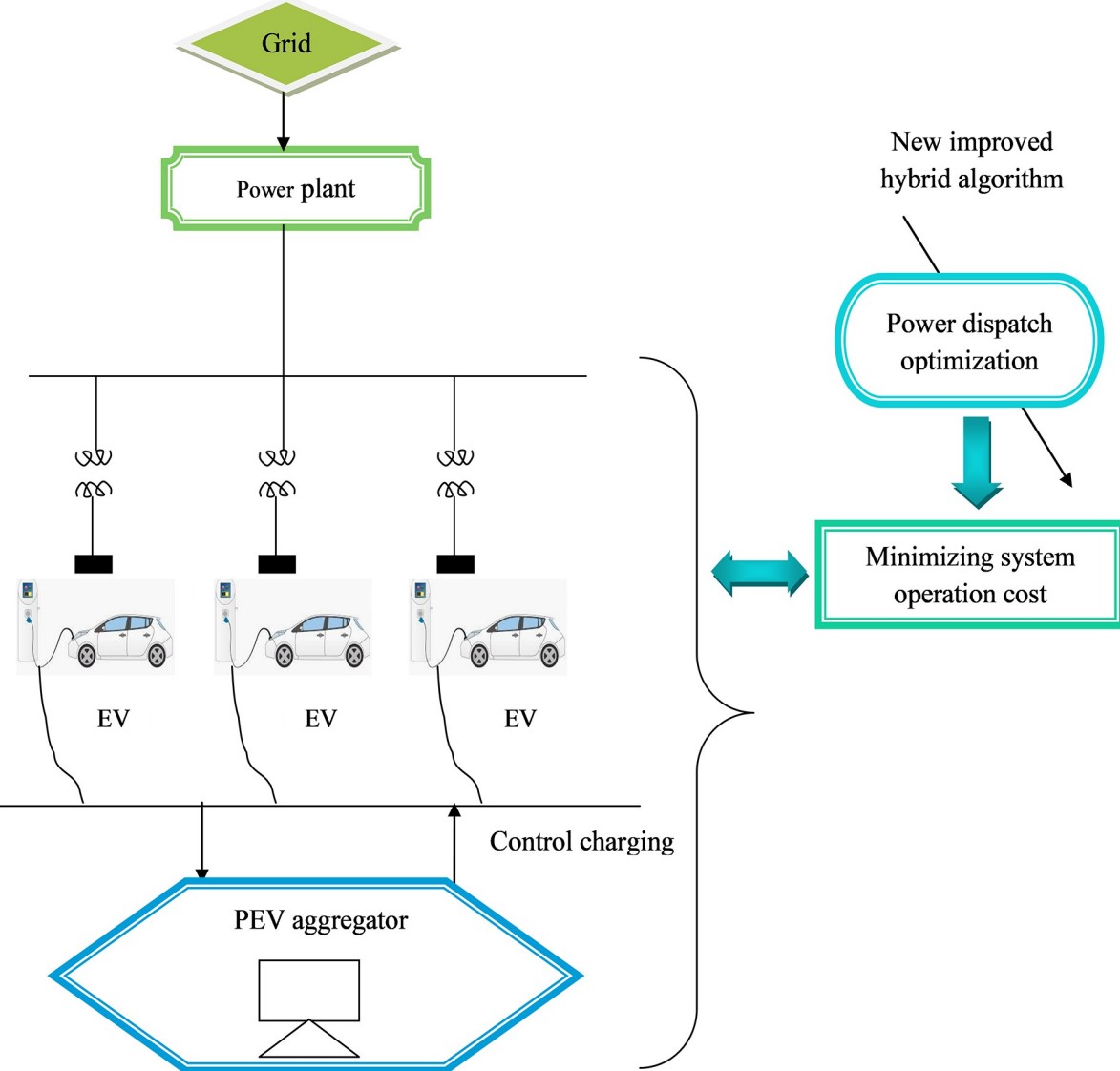

**Fig 1. Working principle of the proposed system.**

optimization process inspired by the way the bird hunts for its prey during the winter and it solves the problems of slow convergence and local optimum trapping. Fig 1 presents the working principle of the proposed optimized charging station system to minimize the time delay taken in case of more number of vehicles to be charged.

The PHEV analysed in these experimental studies is based on a prototype and has a sequential arrangement. An integrated-starter-generator (ISG) and a diesel engine are automatically integrated to form an electric generating unit (EGU) capable of supplying electricity to the battery. The electric motor (EM)shall be used as a driving motor or generator. In reparative braking, the maximum produced power of EM is limited to thirty kilowatts due to battery health. The equal energy efficacy of EGU is computed using the combination of break-specific fuel consumption (BSFC) of the engine and generator efficacy, as depicted using optimal EGU fuel rates. A similar electrical circuit for replicating the lithium-ion phosphate battery design was

described in [5]. The battery strategy's power balance formula is equal to,

$$P_{ib} = P_{tb} + P_{il} = P_{tb} + I^2 R_{ib} \tag{1}$$

where, $P_{ib}$ symbolizes internal battery power, $P_{tb}$ implies terminal battery power, $P_{il}$ represents battery internal power loss, $I$ expresses electrical current, and $R_{ib}$ specifies equivalent internal resistance. The battery dynamics '$B_{SOC}$' is given by,

$$B_{SOC} = F(B_{SOC}) = \frac{-V_{AB} - \sqrt{V^2_{AB} - 4R_{ib}P_{tb}}}{2Q_{ib}R_{ib}} \tag{2}$$

The vehicle level power balance formula for PHEV is equal to:

$$\frac{1}{3600}\left( M_g F_{rr} v_s + \frac{Z_{air}}{21.15} v_s{}^3 + \xi_{in}\frac{dv_s}{dr}v_s = P_{om}\eta_{em}{}^{sgn(P_{om})} \right) \tag{3}$$

$$P_{tb} + P_{EGU} = P_{om} + P_a \tag{4}$$

Here, '$M_g$' refers to mass, $F_{rr}$ refers to rolling resistance coefficient, and $Z_{air}$ refers to air resistance coefficient. $A$ specifies the front area, $v_s$ defines the speed, $\xi_{in}$ demonstrates equal mass inertia, $\eta_{em}$ defines the motor's electric efficiency, and $P_{om}$ specifies the total power invested with the motor or the rehabilitated power during braking. When $P_{om}$ is more than zero (to propel), $sgn$ equals one; when $P_{om}$ is less than zero (to recuperate), $sgn$ equals one. Furthermore, $P_{EGU}$ denotes EGU output power, but $P_a$ denotes power exhausted with auxiliary metrics, namely the braking and electrical steering systems.

## 3. Proposed hybrid AT-AQ optimization algorithm

### 3.1 Objectivefunction and solution encoding

Optimization is the way of assessing all feasible solutions to a function in order to optimize its measurements. The bulk of real-world problems are complex and costly to resolve. As a result, dealing with problems that include large expenditures connected with the installation of charging station motivates rigorous optimization of the charging infrastructure in terms of traffics and the electric grid. The three primary criteria must be addressed in the formulation of the charging station localization dilemma, such as cost, vehicle routing problem (VRP) index, and accessibility index. To address these issues, a meta-heuristic algorithm is designed and developed, which intends to frame the allocation of RES and charging station issues as a multiple-objective approach by including the parameters of the charging stations. As a result, the objective function is given by,

$$F_{obj}(cost) = cost_{fixed} + \min(VRP_{index}, A_{index})$$

$$VRP_{index} = 24 \times Land_{cost} \times Connectorsused(N) \times years(Y) \tag{5}$$

$$A_{index} = Cost_{chargerdevelop} \times (N-1) \times P_{ratedpower\_connector}$$

where, '$cost$' accommodates CS installation and operation costs, '$cost_{fixed}$' is the fixed cost (\$), '$VRP_{index}$' as given in Eq (5) indicates the reliability wherein '$Land_{cost}$' is the rental land cost, 'N' is the number of connectors in charging station, 'Y' is the study time for the particular period of years, '$A_{index}$' reflects the charger developing cost based on charging connector power rating and the number of connectors used for charging, '$Cost_{chargerdevelop}$' specifies the charger development cost, '$P_{ratedpower\_connector}$' is the rated power of the charging connector in

kW. The constraints in respect of the EVs shall be AC power and necessary inverter circuits shall be employed for conversion of AC to DC and the DC power shall be stored in the batteries. Making the EVs into the power grid shall result in voltage drop, energy loss and affecting the peak load of the system. Thus, the constraints in respect of the EVs include,

- Cost of installation of the charging stations

- Increased distribution system power loss

- Difficulty in connecting the EVs for charging directly to the grid

- Problem in the source of electrical energy at a unity power factor and the voltage profile not maintained due to the power system module

- Increased power losses

- Active power loss of the distributed power system network

The above are the specific constraints pertaining to the effective location of charging stations for the Electric Vehicles and all these are handled with the proposed optimization algorithm and this intends to reduce the cost and the losses in the distribution system.

## 3.2 Proposed AT-AQ algorithm

Atom search optimization (ASO) [46] is a physics-inspired meta-heuristic optimization technique inspired by basic molecular dynamics and aimed to solve a variety of optimization problems. The atomic mobility model observable in reality is statistically simulated and mimicked by ASO, wherein atoms combine via interaction force produced by the Lennard-Jones prospective and constraining forces induced by the bond-length potential. It is easy and uncomplicated to employ, and it represents a reasonable alternative to the real-world engineering challenges. Fig 2 provided the detailed flowchart of the AT-AQ model employed in this work for charging station allocation optimization process.

Each atom's location is represented by ASO as a solution, with a heavier mass representing a better solution and vice versa. All atoms in the populace attract or repel each other depending on the exact distance between them, causing the lighter atoms to drift towards the heavier ones. Because larger atoms have a smaller acceleration, they are more inclined to seek effective options in local spaces. Lighter atoms also have more acceleration, allowing them to survey a larger region for new prospective positions. The algorithmic aspects engaged in the formation of the suggested AT-AQ algorithm are also discussed in this section. The conventional mechanics govern atomic motion. Based on Newton's second law, if $F_p$ is the interaction force and $C_p$ is the constraint force acting on the $P^{th}$ atom, and the atom has mass $m_p$, the atom's acceleration becomes,

$$A_p = \frac{F_p + C_p}{m_p} \tag{6}$$

**i). Interaction force.** The stimulating power of atomic motion is the interaction force coming from the $L-J$ potential. In Eq (3), the interaction force acted on the $p^{th}$ atom from the $q^{th}$ atom at the $t^{th}$ iteration may be represented as,

$$U_r\left(R_{pq}\right) = 4\xi\left[\left(\frac{\vartheta}{R_{pq}}\right)^{12} - \left(\frac{\vartheta}{R_{pq}}\right)^{6}\right] \tag{7}$$

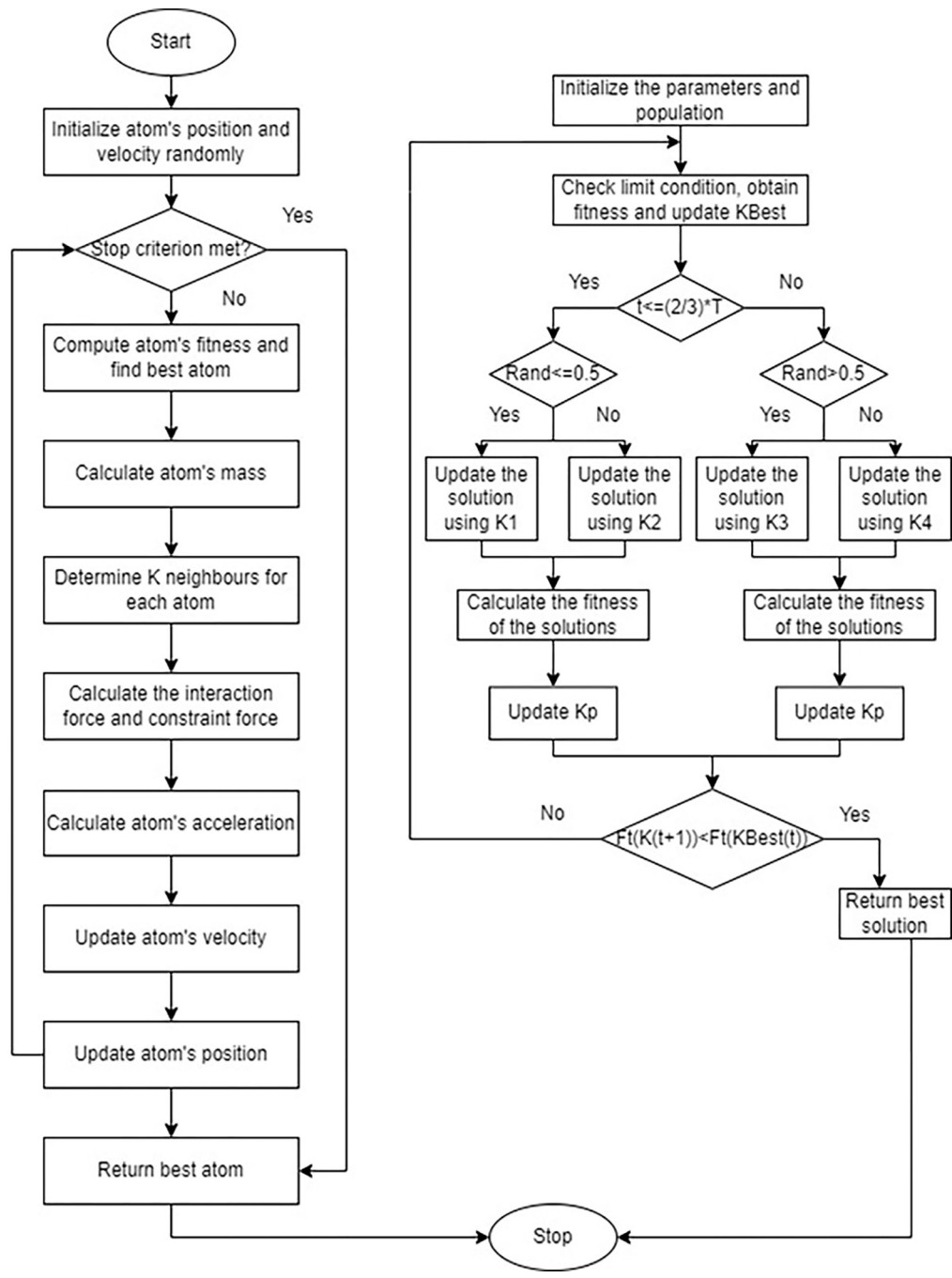

**Fig 2. Flowchart of the proposed AT-AQ model.**

where, $\xi$ is the potential well's depth, $\vartheta$ is the discretized scope at which the inter-particle potential is zero, and $R_{pq} = k_q - k_p$, where $k_p = (k_{p1}, k_{p2}, \ldots, k_{pn})$ and $k_q = (k_{q1}, k_{q2}, \ldots, k_{qn})$ are the $p^{th}$ and $q^{th}$ atom locations in the n-th dimension. As a result, the Euclidean distance between $k_p$ and $k_q$ is given by,

$$R_{pq} = \|k_q - k_p\| = \sqrt{(k_{p1} - k_{q1})^2 + (k_{p2} - k_{q2})^2 + \ldots + (k_{pn} - k_{qn})^2} \qquad (8)$$

The terms $\left(\frac{\vartheta}{R_{pq}}\right)^{12}$ and $\left(\frac{\vartheta}{R_{pq}}\right)^{6}$ represents attraction and repulsion respectively. The interaction force exerted on the $p^{th}$ atom from the $q^{th}$ atom in the $i^{th}$ dimension at the $t^{th}$ time is interpreted by,

$$F_{pq}^{i}(t) = -\nabla U_{r}\left(R_{pq}\right) = \frac{24\xi(t)}{\vartheta(t)}\left[2\left(\frac{\vartheta}{R_{pq}}\right)^{13} - \left(\frac{\vartheta}{R_{pq}}\right)^{7}\right]\frac{R_{pq}}{R_{pq}^{i}} \tag{9}$$

and

$$F_{pq}' = \frac{24\xi(t)}{\vartheta(t)}\left[\left(\frac{\vartheta}{R_{pq}}\right)^{13} - \left(\frac{\vartheta}{R_{pq}}\right)^{7}\right] \tag{10}$$

In Eq (9), the term '$F_{pq}^{i}(t)$' specifies the force exerted from p-th atom to q-th atom, '$R_{pq}$' is the Euclidean distance, '$U_{r}$' specifies the variation metric of the search space. The atoms retain a relative distance from each other that fluctuates over time due to attraction or repulsion, and the deviation in amplitude in repulsion contrasted to equilibrium distance is substantially higher than attraction. Because attraction is positive and repulsion is negative, the atoms cannot converge to a precise point. Eq (10) cannot be utilized adversely for optimization.

$$F_{pq}' = -\eta(t)[(H_{pq}(t))^{13} - (H_{pq}(t))^{7}] \tag{11}$$

where, '$H_{pq}$' specifies the relative Euclidean distance with the minimal and maximal variations as given in Eq (13) and '$\eta(t)$' is the depth task used to alter the repulsion or attraction zone, and it is interpreted as follows:

$$\eta(t) = \alpha_{w}\left(1 - \frac{t-1}{T}\right)^{3}e^{\frac{-20t}{T}} \tag{12}$$

where, $T$ denotes the count of iterations allowed and $\alpha_{w}$ is the depth weight.

$$H_{pq}(t) = \begin{cases} H_{\min} \text{ if } \dfrac{R_{pq}(t)}{\vartheta(t)} < H_{\min} \\[2mm] \dfrac{R_{pq}(t)}{\vartheta(t)} \text{ if } H_{\min} \leq \dfrac{R_{pq}(t)}{\vartheta(t)} \leq H_{\max} \\[2mm] H_{\max} \text{ if } \dfrac{R_{pq}(t)}{\vartheta(t)} > H_{\max} \end{cases} \tag{13}$$

The lower and upper limits of $H_{pq}$ are signified by $H_{min}$ and $H_{max}$, accordingly. $\vartheta(t)$ is the length scale $\sigma(t) = \|k_{pq}(t), \frac{\sum_{J \in K_{Best}}k_{pq}(t)}{K(t)}\|$; where $K_{Best}$ is a subset of $K$ atoms having the best function fitness values.

$$\begin{cases} Hf_{0f_{min}}(d_{f}) \\ Hf_{0f_{max}}(d_{f}) \end{cases} upto\ u_{r} \tag{14}$$

where $u_{r}$ is the upper limit and $d_{f}$ is a drift function that assists the algorithm in devolving

from exploration to exploitation and is given by:

$$d_f(t) = 0.1 \times sin\left(\frac{\pi}{2} \times \frac{t}{T}\right) \tag{15}$$

The total force applied on $p^{th}$ atom from certain atoms is now the weighted sum of the forces in the $i^{th}$ dimension:

$$F_{int}^{i} \sum_{J \in K_{Best}} Rand_q F_{pq}^{i} \tag{16}$$

where, 'F' is the force exerted and the random number in [0, 1] is specified as $Rand_q$.

**ii). Geometric constraints.** The geometric restriction in molecular dynamics is critical in the mobility of atoms. For the sake of versatility, in ASO an atom is supposed to have a covalent bond with the best atom, therefore the best atom exerts a constraint force on each atom. This constraint forces on the $p^{th}$ atom in the $d^{th}$ dimension, which stated as follows:

$$C_{pq}{}^{i} = \lambda(t)(k_{Best}^{i}(t) - k_{p}^{i}(t)) \tag{17}$$

where, '$C_{pq}$' indicates the constraint force and '$\lambda(t)$' is the Lagrangian multiplier is defined as,

$$\lambda(t) = \beta_w e^{\frac{20t}{T}} \tag{18}$$

where, the multiplier weight is designated as $\beta_w$.

**iii). Atomic motion.** Taking interaction force and geometric constraints into account, acceleration of $i^{th}$ atom at a time $t$ in the $i^{th}$ dimension becomes,

$$A_p{}^{i}(t) = \frac{F_p{}^{i}(t) + C_p{}^{i}(t)}{m_p} \alpha_w \left(1 - \frac{t-1}{T}\right)^3 e^{\frac{-20t}{T}} * \sum_{J \in K_{Best}} \frac{Rand_q[2(H_{pq}(t))^{13} - (H_{pq}(t))^7]}{m_p(t)} \frac{(k_p{}^{i}(t) - k_p{}^{i}(t))}{\|k_p(t), k_q(t)\|_2}$$
$$+ \beta_w e^{\frac{-20t}{T}} \frac{F_{Best}{}^{i}(t) - k_p{}^{i}(t)}{m_p(t)} \tag{19}$$

where, $M_p(t)$ represents the mass of $p^{th}$ atom at $t^{th}$ iteration. It is determined as:

$$M_p(t) = e^{-\frac{Ft_p(t) - Ft_{Best}(t)}{Ft_{worst}(t) - Ft_{best}(t)}} \tag{20}$$

$$m_p(t) = \frac{M_p(t)}{\sum_{J=1}^{N} M_q(t)} \tag{21}$$

where, $Ft_{Best}(t) = min_{\{i=1,2,\ldots n\}} Ft_p(t)$ and, $Ft_{worst}(t) = max_{\{i=1,2,\ldots n\}} Ft_p(t)$. The $p^{th}$ atom's position and velocity at a time $(t+1)$ are specified by,

$$V_p{}^{i}(t+1) = Rand_p{}^{i} V_p{}^{i}(t) + A_p{}^{i}(t) \tag{22}$$

$$k_p{}^{i}(t+1) = k_p{}^{i}(t) + V_p{}^{i}(t+1) \tag{23}$$

Exploration ability must be strengthened before it can be used for the optimization issue. As a result, in the early part, each atom must converse with a large number of atoms with higher fitness values as neighbours, hence $K$ must be huge. To improve exploitation near the conclusion of the process, the number of neighbours $K$ must be reduced. As a result, $K$, an

exploration factor is computed as follows:

$$K(t) = N - (N - 2)\sqrt{\frac{t}{T}} \qquad (24)$$

After calculating the mass of an atom, hybridization between ASO and AO is performed.

**iv). Expanded exploration phase.** By high soaring with the vertical stoop in the expanded exploration phase ($K_1$), Aquila finds the prey location and chooses the optimum attacking place. Here, the AO extensively explores the search region from a high altitude to discover where the prey is. This behaviour is statistically represented as in Eq (19),

$$K_1(t + 1) = K_{Best}(t) \times \left(1 - \frac{t}{T}\right) + (K_M(t) - K_{Best}(t) * Rand) \qquad (25)$$

where, $K_1(t+1)$ is the solution of iteration $t$ produced by initial search procedure $K_1$. $K_{Best}(t)$ represents the best solution achieved till the $t^{th}$ iteration, and it represents the fairly accurate location of the prey. The term $\left(1 - \frac{t}{T}\right)$ assist in the regulation of the number of iterations in the expanded search (exploration phase). $K_M(t)$ Signifies the mean of current solutions linked at the $t^{th}$ iteration, as determined by Eq (20). $Rand$ is a number between 0 and 1. The values of $t$ and $T$ represents the current and maximum iteration, respectively.

$$K_M(t) = \frac{1}{N} \sum_{p=1}^{N} K_p(t), \forall q = 1, 2, \ldots, Dim \qquad (26)$$

$$Atompop(p, :) = Simplebounds(Atompop(p, :), low, up) \qquad (27)$$

**v). Narrowed exploration phase.** During the Narrowed exploration phase, when the target zone is located from a higher flight, the Aquila loops over the desired target, approaches the ground, and then strikes. It is regarded as a Contour flying with a brief gliding approach. In this stage, AO closely searches the target prey's chosen region in preparation for the attack. Eq (22) represents this behaviour numerically.

$$K_2(t + 1) = K_{Best}(t) \times Levy(D) + K_R(t) + (y - x) * Rand \qquad (28)$$

where, $K_2(t+1)$ is the response of next iteration of t retrieved by $K_2$, $D$ is the dimension of search space, and Levy($D$) is the levy flight distribution function, which is derived using Eq (23). At the $p^{th}$ iteration, $K_R(t)$ is a random outcome in the range $[1, N]$.

$$Levy(D) = c \times \frac{\mu \times \vartheta}{|v|^{\frac{1}{\tau}}} \qquad (29)$$

where, '$c$' possess a fixed measureof 0.01, $\mu$ and $v$ are random integers ranging from 0 to 1. $\vartheta$ is computed using Eq (30).

$$\vartheta = \left(\Im\left(\frac{1 + \tau \times sine\left(\frac{\pi\tau}{2}\right)}{\Im\left(\frac{1+\tau}{2}\right) \times \tau \times 2^{\left(\frac{\tau-1}{2}\right)}}\right)\right) \qquad (30)$$

where, $\tau$ is a fixed constant measure of 1.5. In Eq (28), the variables $y_s$ and $x_s$ are utilized to

indicate the spiral form in the search process, and they are determined as follows:

$$y_s = R \times cos(\theta) \tag{31}$$

$$x_s = R \times sin(\theta) \tag{32}$$

where, $R = R_0 + u_r{}^* t_0$; $t_0 = 1$: $Dim$; $u_r = 0.0265$; $R_0 = 10$. Also,

$$\theta = -Omega * t_0 * \theta_0 \tag{33}$$

$$\theta_0 = \frac{3 \times \pi}{2} \tag{34}$$

where, $Omega = 0.005$. Also, $R_1$ seizes a value in the range 1 to 20, and $S$ is a minimal value set to 0.00565. $I_1$ is an integer number ranging from 1 to the maximum search space $Dim$, $\omega$ and is a modest measure set to 0.005.

   **vi). Expanded exploitation phase.**   When the prey region is precisely identified and the Aquila is set for landing and attack in the extended exploitation phase ($K_3$), the Aquila stumbles upright with an opening strike to detect the prey response. This technique is known as low flying with a gradual falling attack. AO uses the specified region of prey to come nea rit and perform the hit. Eq (30) represents this behaviour statistically.

$$K_3(t+1) = (K_{Best}(t) - K_M(t)) \times \alpha - Rand + ((U_rB - L_rB) \times Rand + L_rB) \times \delta \tag{35}$$

where, $K_3(t+1)$ indicates the outcome of the following iteration of $t$ produced by the third search mechanism ($K_3$). The approximate position of prey till the $p^{th}$ iteration (the best-obtained solution) is denoted by $K_{Best}(t)$, and the mean of current solution at the $t^{th}$ iteration is denoted by $K_M(t)$, and is found via Eq (26). $Rand$ is a number varying in the limit of 0 to 1. $\alpha$ and $\delta$ are adjustment parameters of exploitation, and in this work $\alpha$ and $\delta$ are set to a low measure of 0.1. The lower limit is denoted by $L_rB$, while the upper bound is denoted by $U_rB$ in the following problem.

   **vii). Narrowed exploitation phase.**   When the Aquila gets close to the victim during the Narrowed exploitation phase (X4), the Aquila strikes the victim using stochastic action. This technique is known as stroll and grabs prey. At the last step, the prey is attacked by the Aquila at the final spot. Eq (36) represents this behaviour precisely.

$$K_4(t+1) = Q_F \times K_{Best}(t) - (\mathfrak{R}_1 \times K(t) \times Rand) - \mathfrak{R}_2 \times Levy(D) + Rand \times \mathfrak{R}_1 \tag{36}$$

where $K_4(t+1)$ specifies the outcome of the fourth search method's next iteration $t$. $Q_F$ is a quality function that assist in balancing the search methods and is determined using Eq (37).

$$Q_F(t) = t^{\frac{2 \times Rand - 1}{(1-T)^2}} \tag{37}$$

   $\mathfrak{R}_1$ indicates multiple AO movements utilized to monitor the prey during the chase, which are created by Eq (38). $\mathfrak{R}_2$ displays decreasing measures from 2 to 0, indicating the flight slope of AO utilized to track the food at the time of trip from $first(1)$ to $last(t)$ position, as calculated via. Eq (39). At the $t^{th}$ iteration, the present solution is $K(t)$. The terms $\mathfrak{R}_1$ and $\mathfrak{R}_2$ are expressed as,

$$\mathfrak{R}_1 = 2 * Rand() - 1 \tag{38}$$

$$\mathfrak{R}_2 = 2 * (1 - (Itr / max\,I\,tr)) \tag{39}$$

$Q_F$ is a quality function that assist in balancing the search methods and is determined using Eq (40).

$$Q_F = max \, I \, tr^{\wedge}((2*Rand() - 1/(1 - max \, I \, tr)^{\wedge}2) \tag{40}$$

The pseudo code of the developed AT-AO framework is presented below in Algorithm 1 and its flowchart is depicted in Fig 2.

```
Algorithm 1: Proposed AT-AQ Algorithm
Arbitrarily initialize a set of atoms. K (Solutions)and their velocity
V, and Ft_Best = Inf.
While the stop criterion is not satisfied do
  For each atom K_p do
    Compute the fitness value Ft_p;
    If Ft_p<Ft_Best then
      Ft_Best = Ft_p;
      K_Best = K_p;
    End if.
    Evaluate the mass using Eq (20) and Eq (21);
    Employing Eq (24), compute its K neighbors.
    Utilizing Eq (16) and Eq (17), estimate the interaction force F_p
and the constraint force correspondingly.
    Enumerate acceleration via Eq (19);
    Modernize velocity via Eq (22);
    Modernize position via Eq (23);
  End for.
End while.
Evaluate best solution so far K_Best
Initialize population K of the AO.
Initialize parameters.
While
  Stopping condition is not satisfied
    do
  Estimate fitness measures
  K_Best(t) = Determinethebestobtainedsolution
accordingtothefitnessvalues.
  for (i = 1,2...,N)
    do
  Modernize mean of current solution K_M(t)
  Modernize x_s, y_s, ℜ_1, ℜ_2, Levy(D), etc.
    if t ≤ (2/3)*T then
      Modernize ℜ_2 and ℜ_1 use Eq (38) and Eq (39) correspondingly.
      If Rand≤0.5 then
        Modernize current solution via Eq (25)
        If Ft(K_1(t+1))<Ft(K(t)) then
          K(t) = K_1(t+1)
          If Ft(K_1(t+1))<Ft(K_Best(t)) then
            K_Best(t) = K_1(t+1)
          End if
        End if
        Else
        Modernize current solution via Eq (28).
        If Ft(K_2(t+1))<Ft(K(t)) then
          K(t) = K_2(t+1)
          If Ft(K_2(t+1))<Ft(K_Best(t)) then
            K_Best(t) = K_1(t+1)
          End if
        End if
```

```
        End if
        Else
        If Rand>0.5 then
          Modernize current solution via Eq (35).
          If Ft(K₃(t+1))<Ft(K(t)) then
            K(t) = K₃(t+1)
            If Ft(K₃(t+1))<Ft(K_Best(t)) then
              K_Best(t) = K₃(t+1)
            End if
          End if
          Else
          Modernize current solution via Eq (36).
          If Ft(K₄(t+1))<Ft(K(t)) then
            K(t) = K₄(t+1)
            If Ft(K₄(t+1))<Ft(K_Best(t)) then
              K_Best(t) = K₄(t+1)
            End if
          End if
        End if
      End if
    End for
  End while
  Return K_Best
```

## 4. Results and analysis

### 4.1 Simulation process

The proposed CS allocation system was implemented and analysed in MATLAB environment. As a result, the proposed AT-AQ based model's cost and time were compared to those of other traditional schemes such as Monkey Search Algorithm (MS) [47], Genetic Algorithm (GA) [48], Aquila Optimizer (AQ) [49], and Atom Search Algorithm (AT) [49]. The analysis was carried out in two different scenarios. Different charging levels were assigned for 24 hours in the first scenario, and in the next scenario. Initially, the cost of charging a CS is held at zero time intervals and the charging cost varies with varied time intervals while charging. The performance analysis was done with time intervals of 0, 5, 10, 15, 20, and 25 hours. Furthermore, the analysis was conducted with regard to "Best, worst, mean, median and standard deviation (STD)" to express the superiority of the AT-AQ model. The evaluation of the methods is carried out by comparing the proposed algorithm with several state-of-art algorithms concerning total cost, voltage profile and so on.

### 4.2 Performance analysis

The dataset employed to test the proposed optimization model is the usage of electric vehicles in the campus of Georgia Tech, Atlanta, USA and the vehicles were charged at the conference centre parking station and 150 vehicles were flying around the campus. The mean driving distance of the vehicles is 31 km. The regional distribution will be around the campus of Georgia Tech, Atlanta, USA in this research study and the datasets is as in reference [50]. Performance analysis curves are displayed along the x and y axes for conventional cost and time values. The charging cost of multiple scenarios was shown by the characteristics curve at varied time intervals of 0, 5, 10, 15, 20, and 25. Three alternative situations may be used to determine the total cost necessary for a specific time period of the CS allocation model. The cost of the models was taken as Total cost, C1 Cost, and C2 Cost for case 1 and Case 2 with three different scenarios,

such as Scenario 1, Scenario 2, and Scenario 3, correspondingly. Three scenarios have been employed for this study, the scenarios include,

Scenario 1: Optimal placement based on distribution system conjunction with transportation system

Scenario 2: Optimal placement based on distribution generators with previous optimal charging load

Scenario 3: Allocation of distribution generators and charging station in distribution system optimally based on earlier optimal load

As a result, the performance of the characteristics curve was depicted for relevant cost values in the range of $10^{11}$ for Case 1, $10^8$ Case 2 and $10^8$ for Case 3. More specifically, the proposed approach has needed lower charging costs than the earlier models. That is, the developed charging station allocation model outperforms traditional models, such as MS, GA, AQ, and AT. Furthermore, the analysis was carried out in terms of "best, worst, mean, median, and STD" to demonstrate the efficiency of the AT-AQ model. The performance measurements demonstrate how effective the CS model is based on the suggested work. As a result, the implemented AT-AQ technique has achieved a low-cost value for specific time periods. This research demonstrates the superiority of the proposed AT-AQsystem in charging the CS model. Tables 2–4 reveals the effectiveness of the proposed system's "C1 Cost" over the conventional schemes by best, worst, mean, median and STD parameters forcase 1, case 2, and case 3accordingly.

**Table 2. Scenario 1.**

|  | MS [47] | GA [48] | AQ [49] | AT [49] | Proposed AT-AQ optimizer |
|---|---|---|---|---|---|
| Best (e+11) | 1.227 | 1.2272 | 1.2272 | 1.2274 | 1.2271 |
| Worst (e+11) | 2.5493 | 2.5487 | 2.5493 | 2.5491 | 2.4451 |
| Mean (e+11) | 1.8034 | 1.9029 | 1.8756 | 1.9869 | 1.6131 |
| Median(e+11) | 2.07 | 2.07 | 2.07 | 2.2389 | 1.2278 |
| STD(e+11) | 0.55882 | 0.50724 | 0.57967 | 0.56821 | 0.52029 |

**Table 3. Scenario 2.**

|  | MS [47] | GA [48] | AQ [49] | AT [49] | Proposed AT-AQ optimizer |
|---|---|---|---|---|---|
| Best (e+11) | 1.227 | 1.2271 | 1.2273 | 1.2271 | 1.2272 |
| Worst(e+11) | 2.5476 | 2.5482 | 2.5492 | 2.5497 | 2.5489 |
| Mean(e+11) | 1.7104 | 1.914 | 1.8892 | 1.9603 | 1.713 |
| Median(e+11) | 1.2281 | 2.0713 | 2.0712 | 2.2394 | 1.2277 |
| STD(e+11) | 0.55057 | 0.5159 | 0.54485 | 0.59209 | 0.55258 |

**Table 4. Scenario 3.**

|  | MS [47] | GA [48] | AQ [49] | AT [49] | Proposed AT-AQ optimizer |
|---|---|---|---|---|---|
| Best(e+11) | 1.227 | 1.2271 | 1.2273 | 1.2272 | 1.2273 |
| Worst(e+11) | 2.5492 | 2.549 | 2.5483 | 2.5495 | 2.549 |
| Mean(e+11) | 1.8989 | 1.7317 | 1.8535 | 1.9286 | 1.8377 |
| Median(e+11) | 2.1548 | 1.6489 | 2.2387 | 2.071 | 2.07 |
| STD(e+11) | 0.54688 | 0.53683 | 0.55042 | 0.53209 | 0.59366 |

**Table 5. Scenario 1.**

|  | MS [47] | GA [48] | AQ [49] | AT [49] | Proposed AT-AQ optimizer |
|---|---|---|---|---|---|
| Best(e+8) | 1.9904 | 1.9915 | 1.9907 | 1.9903 | 1.9881 |
| Worst(e+8) | 2.4351 | 2.4403 | 2.095 | 2.2225 | 1.9889 |
| Mean(e+8) | 2.2139 | 2.2056 | 2.0356 | 2.0141 | 1.9884 |
| Median(e+8) | 2.228 | 2.1887 | 2.0429 | 2.0013 | 1.9884 |
| STD(e+8) | 0.14264 | 0.13299 | 0.3324 | 0.48252 | 0.00019 |

**Table 6. Scenario 2.**

|  | MS [47] | GA [48] | AQ [49] | AT [49] | Proposed AT-AQ optimizer |
|---|---|---|---|---|---|
| Best(e+8) | 2.0237 | 2.0099 | 1.9968 | 1.9911 | 1.9881 |
| Worst(e+8) | 2.4224 | 2.4274 | 2.4213 | 2.1609 | 1.9887 |
| Mean(e+8) | 2.23 | 2.2211 | 2.2002 | 2.0483 | 1.9884 |
| Median(e+8) | 2.2342 | 2.2323 | 2.1845 | 2.041 | 1.9885 |
| STD(e+8) | 0.11681 | 0.11359 | 0.013074 | 0.044462 | 0.00015 |

**Table 7. Scenario 3.**

|  | MS [47] | GA [48] | AQ [49] | AT [49] | Proposed AT-AQ optimizer |
|---|---|---|---|---|---|
| Best(e+8) | 2.0418 | 1.9943 | 1.9883 | 1.998 | 1.9896 |
| Worst(e+8) | 2.4261 | 2.425 | 1.9888 | 2.394 | 2.0125 |
| Mean(e+8) | 2.2136 | 2.2137 | 1.9886 | 2.1062 | 1.9973 |
| Median(e+8) | 2.2238 | 2.2052 | 1.9886 | 2.0624 | 1.9977 |
| STD(e+8) | 0.11544 | 0.13678 | 0.000118 | 0.10933 | 0.0058 |

**Table 8. Scenario 1.**

|  | MS [47] | GA [48] | AQ [49] | AT [49] | Proposed AT-AQ optimizer |
|---|---|---|---|---|---|
| Best(e+11) | 1.229 | 1.2293 | 1.2293 | 1.2294 | 1.2291 |
| Worst(e+11) | 2.5515 | 2.551 | 2.5513 | 2.5511 | 2.4471 |
| Mean(e+11) | 1.8056 | 1.914 | 1.8892 | 1.9603 | 1.713 |
| Median(e+11) | 2.0724 | 2.0732 | 2.0725 | 2.2409 | 1.2298 |
| STD(e+11) | 0.55879 | 0.50721 | 0.57966 | 0.56821 | 0.52029 |

Tables 5–7 reveals the effectiveness of the proposed system's "C2 Cost" over the conventional schemes by best, worst, mean, median and STD parameters for case 1, case 2, and case 3 accordingly.

Tables 8–10 reveals the effectiveness of the proposed system's "total cost" over the conventional schemes by best, worst, mean, median and STD parameters for case 1, case 2, and case 3 accordingly.

The statistics precisely demonstrated that the proposed charging strategy outperforms the standard parameters in terms of performance. The newly modelled AT-AQ optimizer outperforms the MS [47], GA [48], AQ [49], and AT [49] models in terms of the best solution. Furthermore, it achieved best, worst, mean, median, and standard deviation than conventional systems. As a result, the suggested strategy is more effective than the existing optimization techniques.

**Table 9. Scenario 2.**

|  | MS [47] | GA [48] | AQ [49] | AT [49] | Proposed AT-AQ optimizer |
|---|---|---|---|---|---|
| Best(e+11) | 1.2293 | 1.2293 | 1.2295 | 1.2292 | 1.2292 |
| Worst(e+11) | 2.5499 | 2.5503 | 2.5513 | 2.5517 | 2.5508 |
| Mean(e+11) | 1.7126 | 1.9163 | 1.8914 | 1.9623 | 1.715 |
| Median(e+11) | 1.2302 | 2.0736 | 2.0733 | 2.2415 | 1.2297 |
| STD(e+11) | 0.55056 | 0.51591 | 0.54481 | 0.59209 | 0.55258 |

**Table 10. Scenario 3.**

|  | MS [47] | GA [48] | AQ [49] | AT [49] | Proposed AT-AQ optimizer |
|---|---|---|---|---|---|
| Best(e+11) | 1.229 | 1.2294 | 1.2293 | 1.2293 | 1.2293 |
| Worst(e+11) | 2.5515 | 2.5514 | 2.5503 | 2.5518 | 2.551 |
| Mean(e+11) | 1.9012 | 1.7339 | 1.8555 | 1.9307 | 1.8397 |
| Median(e+11) | 2.157 | 1.6513 | 2.2407 | 2.0732 | 2.0721 |
| STD(e+11) | 0.54689 | 0.53681 | 0.55042 | 0.53209 | 0.59366 |

Figs 3A–5B exhibits the charging station allocation model's charging cost at varying time transients during the first case for scenario 1, 2, and 3 respectively. Also, 3b, 4b, and 5b are the convergence plots for the plots 3a, 4a, and 5a respectively. The performance curve is presented here for various strategies with costs of 1.2, 1.4, 1.6, 1.8, 2.0, 2.2, 2.4, and 2.6 with the time periods of 0, 5, 10, 15, 20 and 25. The graphs proved that the proposed model charged faster than the existing models, requiring a lower charging cost.

Figs 6A–8B shows the CS allocation model's charging cost at varying time transients during the second case for scenario 1, 2, and 3, respectively. Here, 6b, 7b, and 8b are the convergence plots for 6a, 7a, and 8a respectively. The performance curve is presented here for various strategies with costs of 2, 2.05, 2.1, 2.15, 2.2, 2.25, 2.3, 2.35, 2.4, and 2.45 with the intervals of 0, 5,

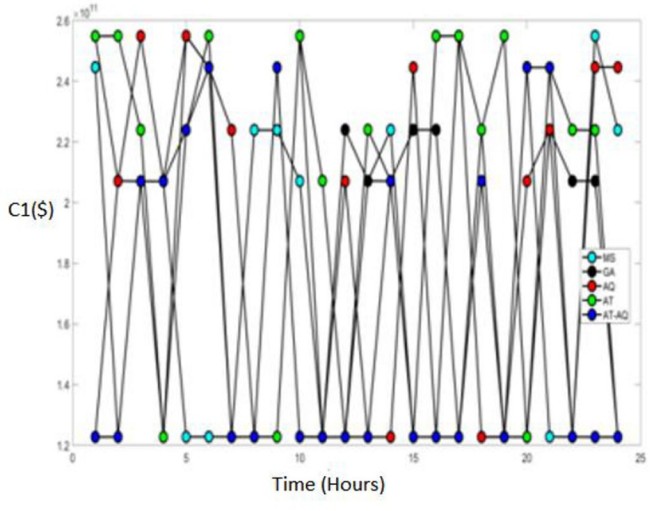

a

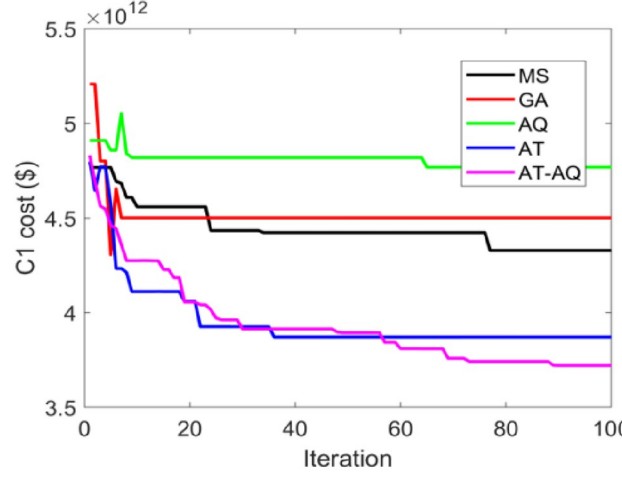

b

**Fig 3.** a Cost 1 Vs Time Scenario 1. b Convergence plot Scenario 1b.

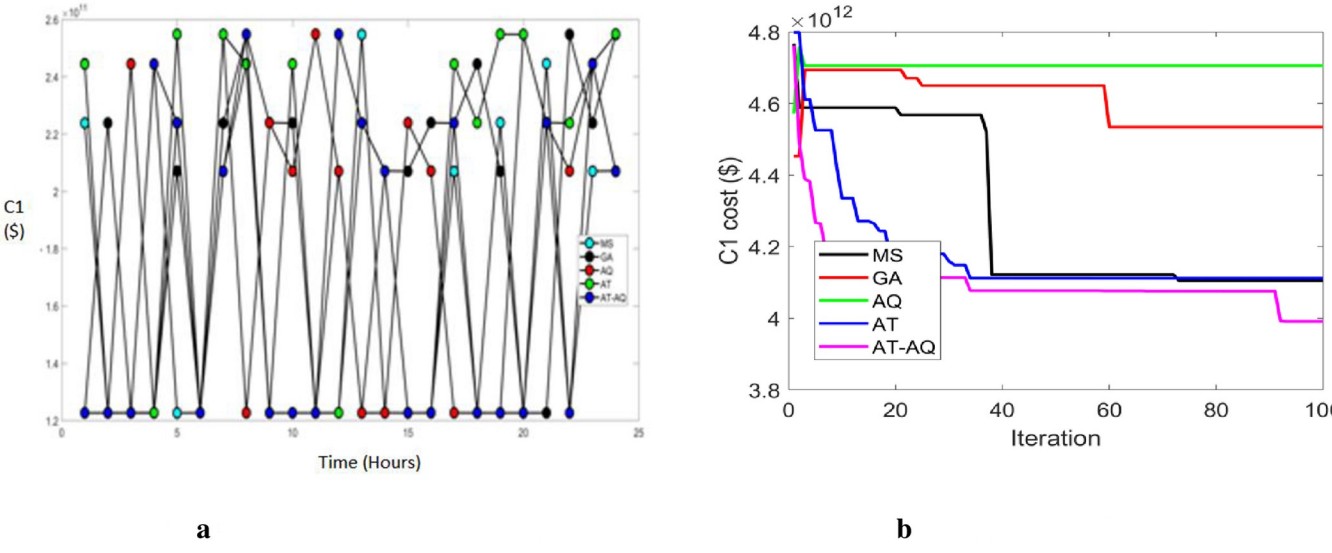

**Fig 4.** a Cost 1 Vs Time Scenario 2. b Convergence plot Scenario 2.

10, 15, 20 and 25. The graphs revealed that the proposed model charged faster than the existing models, requiring a lower charging cost.

Figs 9–11 showcase the Reserve cost of the diesel fuel engine with varying time instants for scenario 1, 2, and 3, respectively. From the graphs, it is clearly depicted that the proposed model has attained improved efficacy than the conventional schemes such as MS, GA, AQ, and AT optimizations.

Figs 12A–14B depict the total cost of the system with varying time intervals for scenario 1, 2, and 3, respectively. Also, 12b, 13b and 14b are the convergence plots for 12a, 13a and 14a respectively. Here the total cost of the proposed system is obtained by evaluating the sum of both the cases. i.e; *Total cost* = $C_1 Cost + C_2 Cost$. Moreover, the above graphs clearly depicted

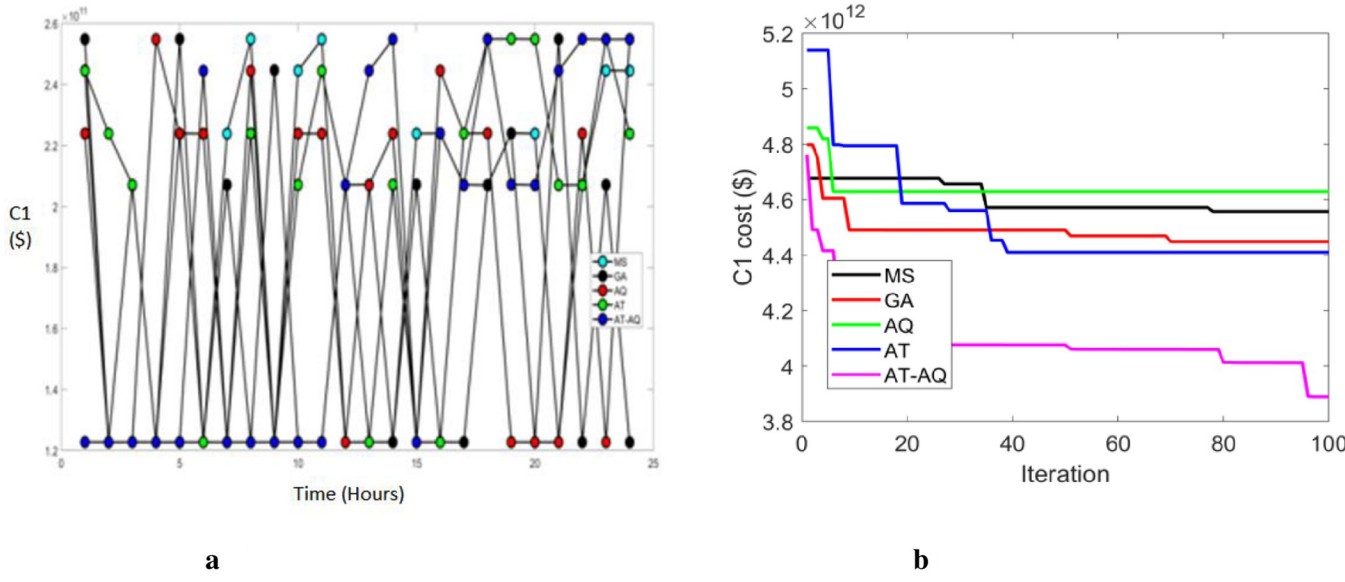

**Fig 5.** a Cost 1 Vs Time Scenario 3. b Convergence plot Scenario 3.

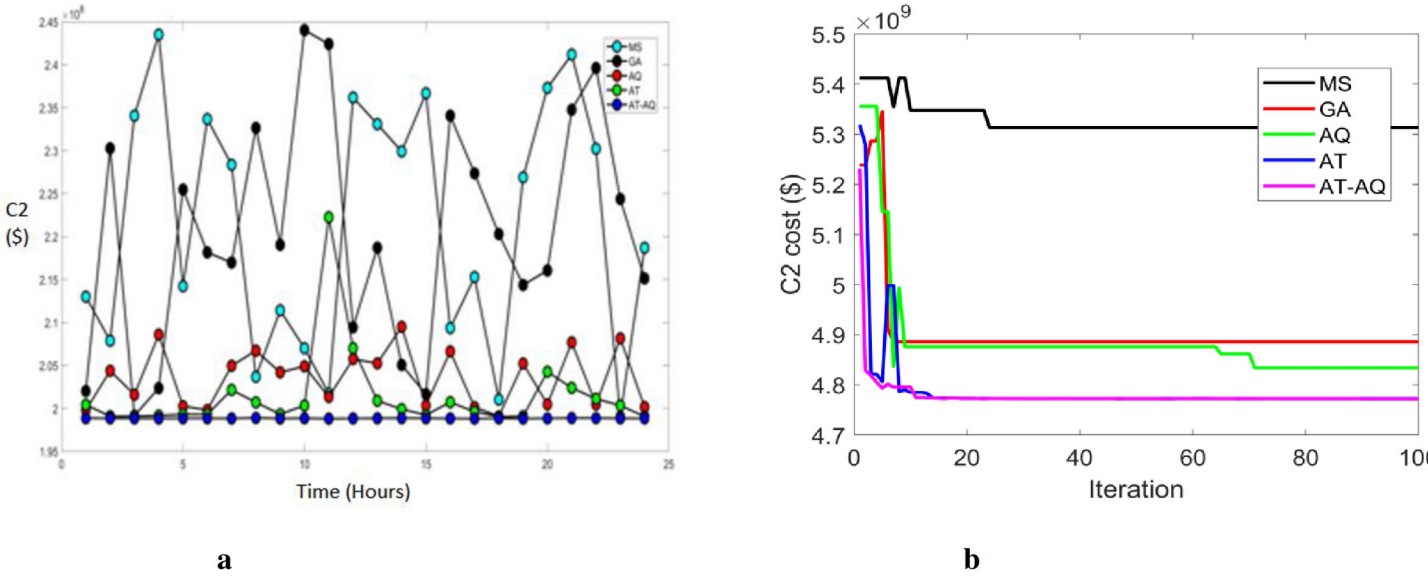

**a** **b**

**Fig 6.** a Cost 2 Vs Time Scenario 1. b Convergence plot Scenario 1.

that the proposed framework has a faster charging capacity than the classical schemes and hence required lower costs. Also, the proposed scheme converges more efficiently than the existing schemes.

On performing the necessary simulation process employing the proposed AT-AQ optimization algorithm, it has been observed that the cost incurred during the charging mechanism has significantly been reduced. The total cost incurred were evaluated for the two costs $C_1$ and $C_2$ and for the total cost as well. For the considered three scenarios, the variation in the

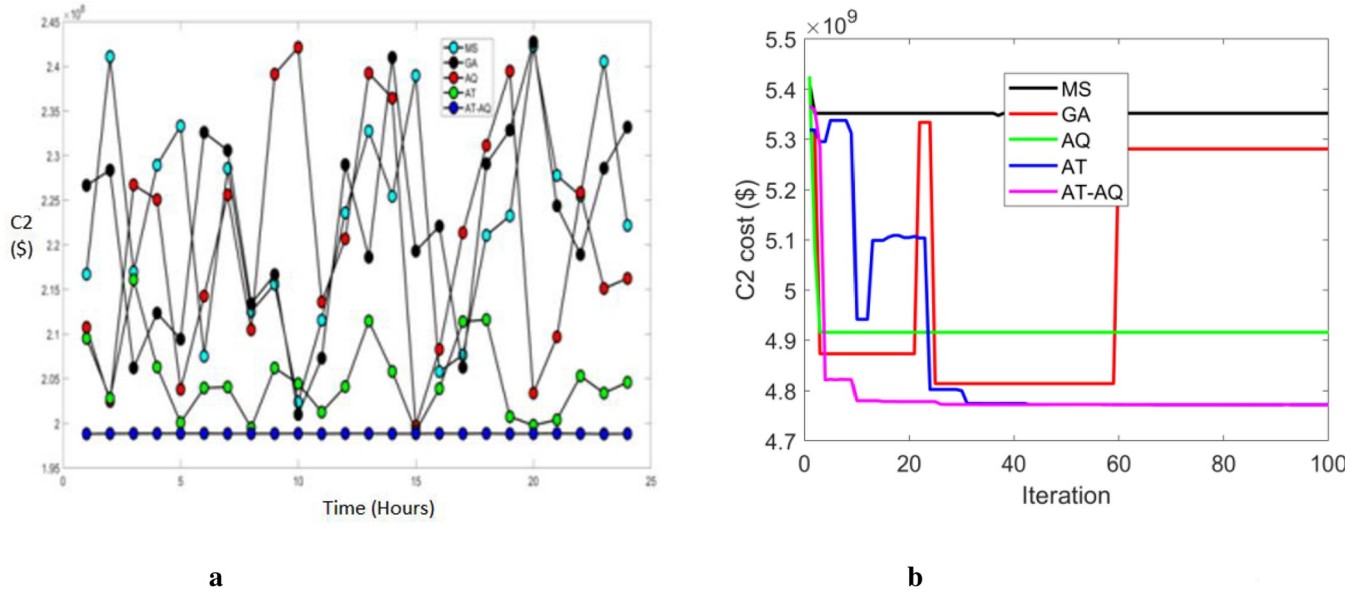

**a** **b**

**Fig 7.** a Cost 2 Vs Time Scenario 2. b Convergence plot Scenario 2.

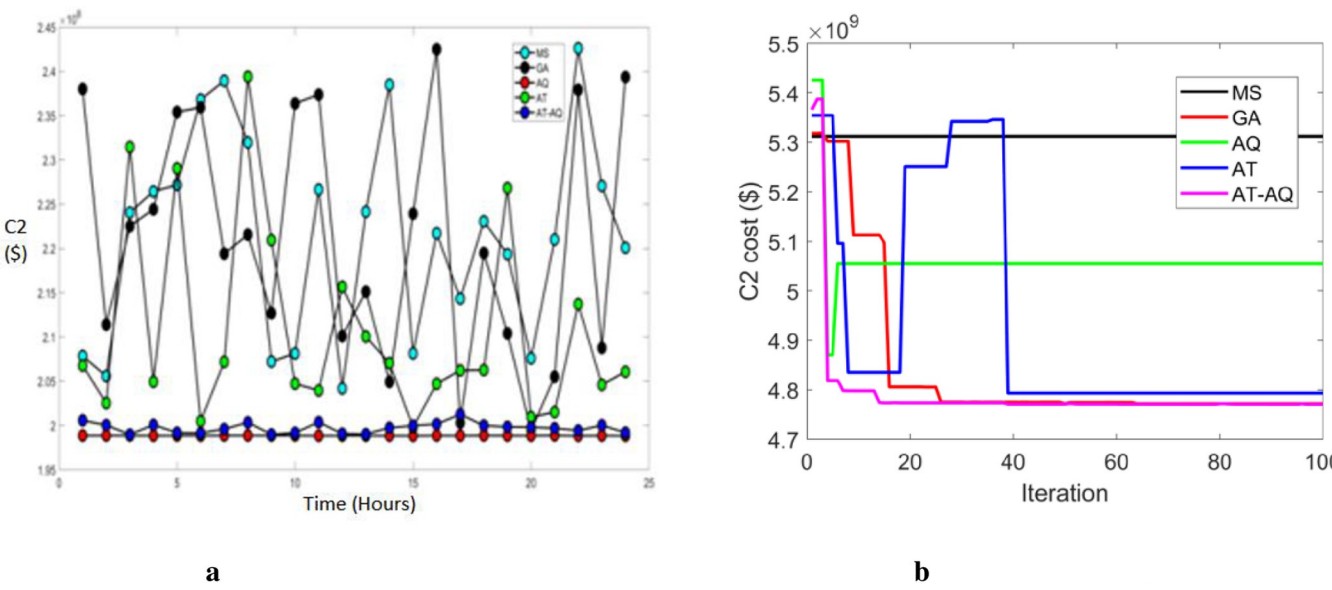

**a**
 
**b**

**Fig 8.** a Cost 2 Vs Time Scenario 3. b Convergence plot Scenario 3.

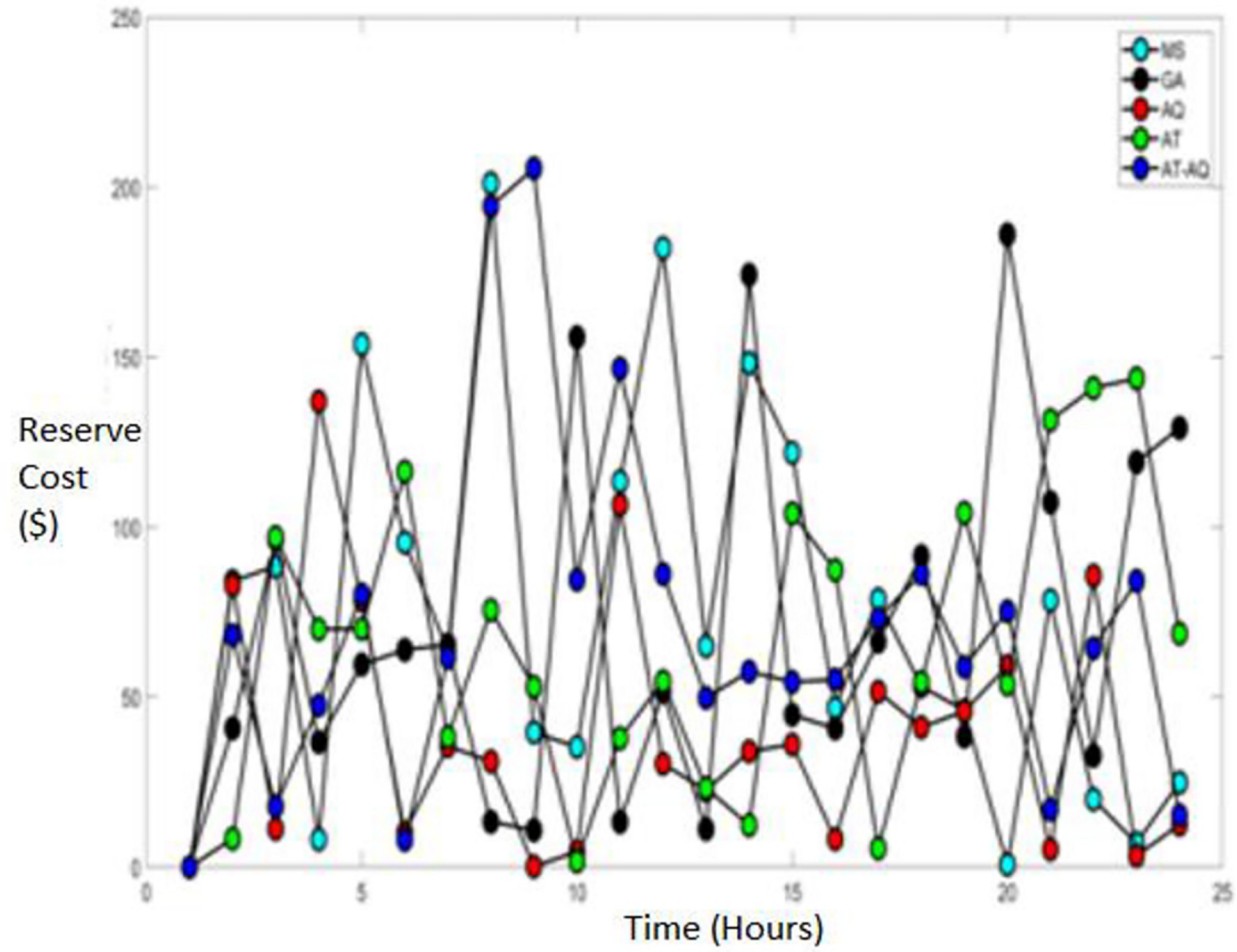

**Fig 9. Reserve cost Scenario 1.**

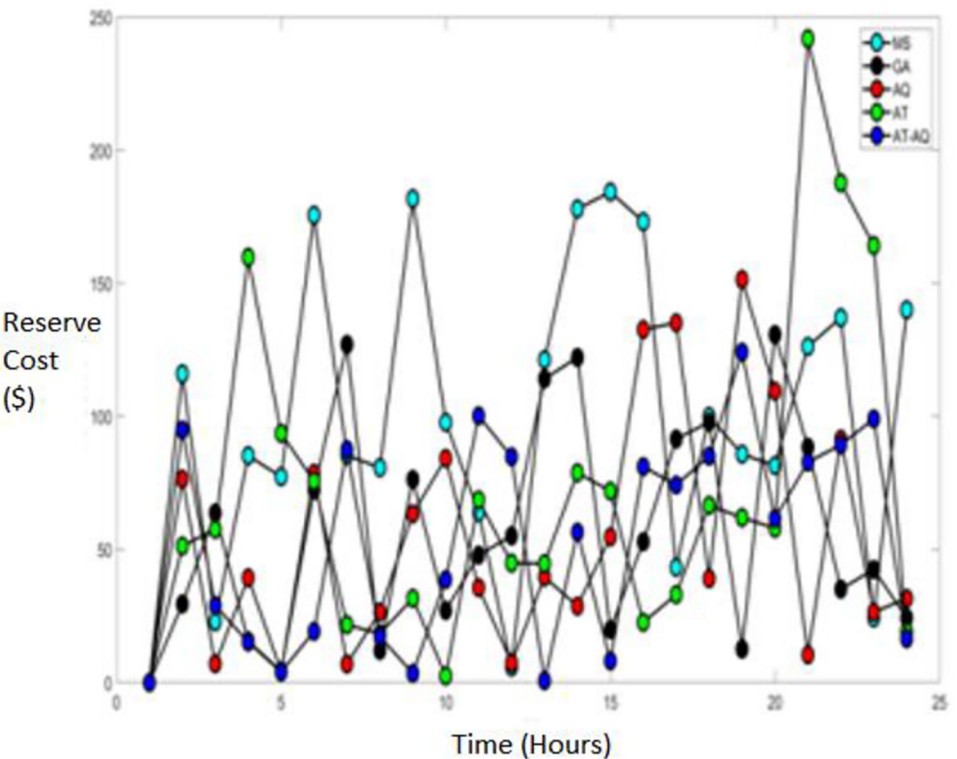

**Fig 10. Reserve cost Scenario 2.**

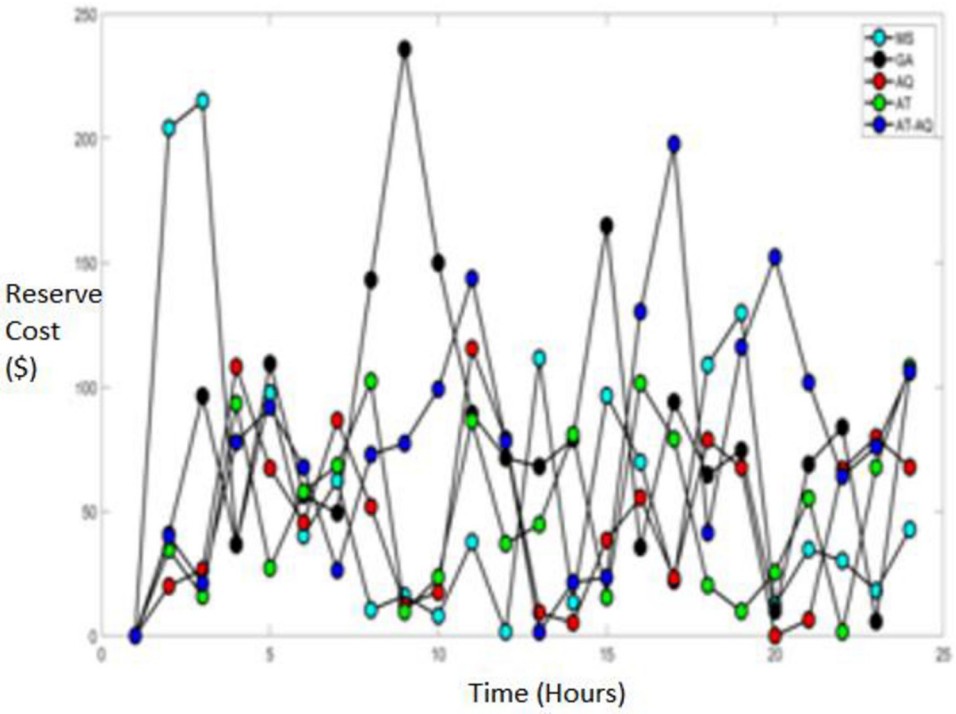

**Fig 11. Reserve cost Scenario 3.**

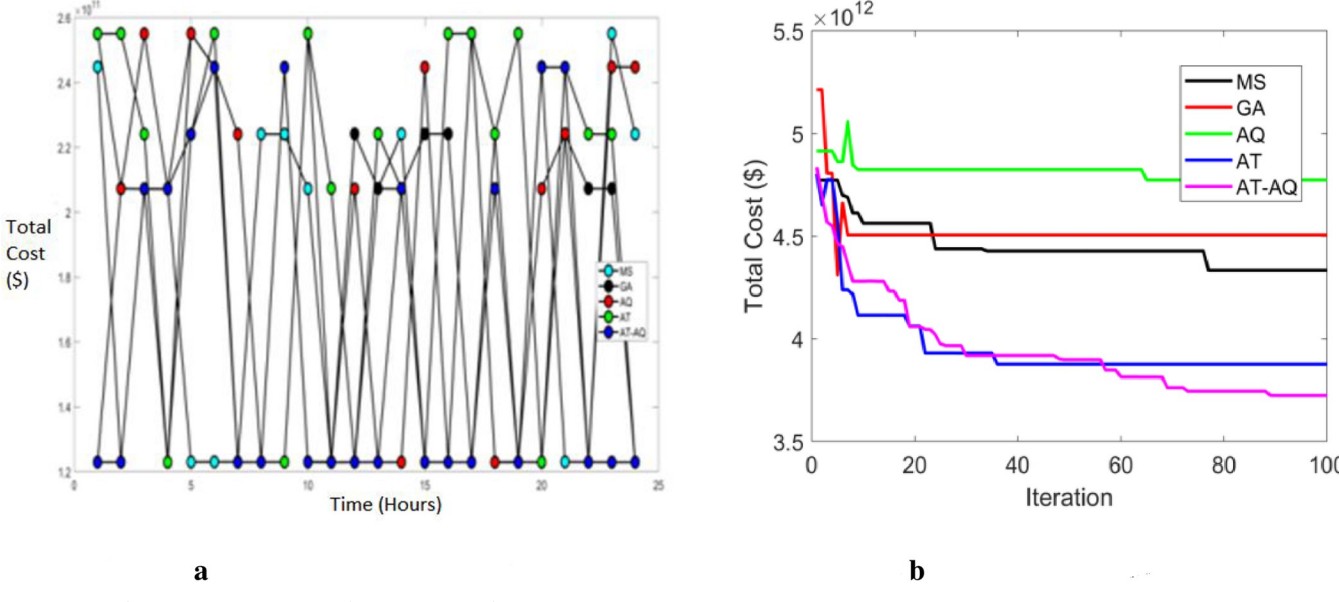

**Fig 12.** a Total cost Vs Time Scenario 1. b Convergence Plot Scenario 1.

standard deviation evaluated using simulation is 0.52029, 0.55258 and 0.59366. In respect of the cost C2, it has been noted that the standard deviation parameter includes 0.00019, 0.00015 and 0.0058. The median variation of the total cost is observed to be 1.2298, 1.2297 and 2.0721 for the scenarios respectively. It confirms that the cost gets minimized to the most possible extent proving the versatility of the proposed AT-AQ optimization algorithm in respect of improvement in the exploration and exploitation mechanism. The force exerted and the update mechanism results in attainment of global stability with minimized cost-effective value.

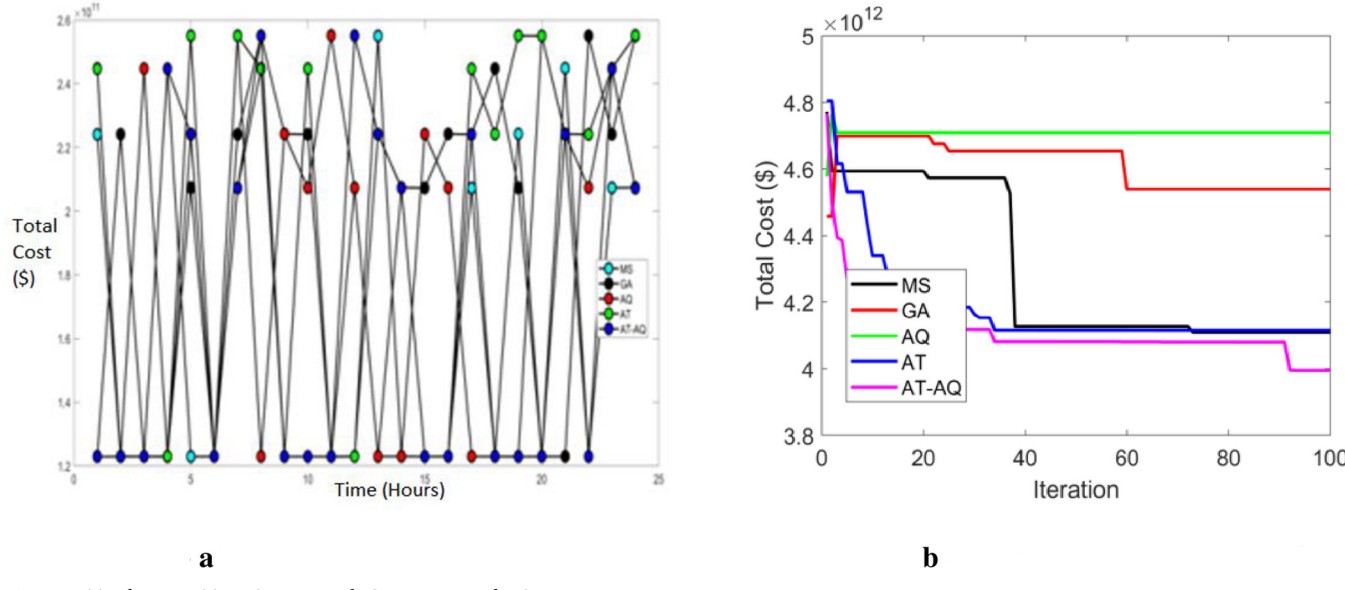

**Fig 13.** a Total cost Vs Time Scenario 2. b Convergence Plot Scenario 2.

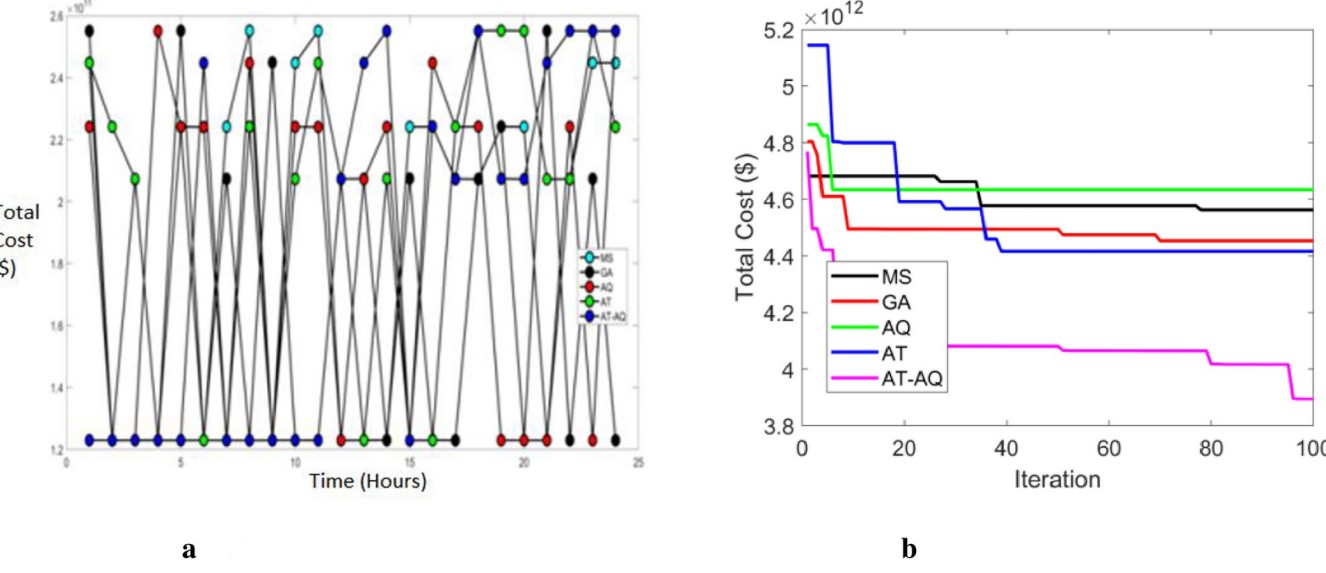

**Fig 14.** a Total cost Vs Time Scenario 3. b Convergence Plot Scenario 3.

## 5.Conclusion

A systematic optimization approach of charging electric vehicles (EVs) as well as allocating a better place for their installations has been presented in this research study. The main destination purpose is to establish a charging network at an affordable cost while maintaining the distribution network's operating qualities. To fulfill the consequences of the variables, these challenges are handled using meta-heuristic algorithms and optimum planning based on renewable energy sources. As a result, this study proposed a novel strategy to conceptualize the distribution of RES and charging station problems as a multiple-objectives strategy by including charging station characteristics. A novel optimizer called Atom Search Woven Aquila Optimization Algorithm (AT-AQ) that incorporates the concepts of both Aquila Optimizer (AO) and Atom Search Optimization (ASO) Algorithms was devised and utilized to execute the charging station distribution process. The effectiveness of the proposed method has been analyzed with respect toBest, Worst, Mean, Median, and STD parameters in the charging stations, and the results thus obtained are compared with the conventional strategies.

Significantly, the total cost incurred using the proposed technique for the three scenarios includes, 1.713, 1.715 and 1.8397 respectively proving the efficacy of the approach. The modelled optimizer has been superior to other existing strategies in terms of its charging efficiency in a short duration of time at a lower cost. When comparing the cost efficiency of the proposed AT-AQ model to the conventional MS, GA, AQ and AT strategies, the proposed AT-AQ model has an enhanced best solution of 1.2271e+11 in case 1 and 1.9881e+08 in case 2, which is better than the existing MS, GA, AQ and AT models. A comprehensive set of simulations was done to express the achievement of the suggested methodology, and the outcomes were evaluated using various metrics. The experimental outcomes demonstrated that the proposed framework outperformed the other conventional strategies. The method also offers strong and effective steady-state performance and rapid dynamic response in respect of the charging station allocation.

## Data availability statement

- Author generated code shall be made available and it can be reused and well documented

- Data is from the public dataset as given -www.kaggle.com/datasets/claytonmiller/campus-electric-vehicle-charging-stations-behavior/metadata and is included in the suporitng information file.

## Supporting information

**S1 File.**
(ZIP)

**S2 File.**
(CSV)

**S3 File.**
(XLSX)

**S4 File.**
(XLSX)

**S5 File.**
(XLSX)

## Author Contributions

**Conceptualization:** Ayyappan Subramaniam.

**Data curation:** Ayyappan Subramaniam.

**Formal analysis:** Ayyappan Subramaniam.

**Funding acquisition:** Ayyappan Subramaniam.

**Investigation:** Ayyappan Subramaniam, Lal Raja Singh Ravi Singh.

**Methodology:** Ayyappan Subramaniam.

**Project administration:** Ayyappan Subramaniam.

**Resources:** Ayyappan Subramaniam.

**Software:** Ayyappan Subramaniam.

**Supervision:** Ayyappan Subramaniam, Lal Raja Singh Ravi Singh.

**Validation:** Ayyappan Subramaniam.

**Visualization:** Ayyappan Subramaniam.

**Writing – original draft:** Ayyappan Subramaniam.

**Writing – review & editing:** Ayyappan Subramaniam.

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
