## [Decision Letter · Decision Letter 0]

15 Dec 2022

PONE-D-22-29934Optimal Planning and Allocation of Plug-in Hybrid Electric Vehicles Charging Stations using a Novel Hybrid Optimization Technique PLOS ONE

Dear Dr. Subramaniam,

Thank you for submitting your manuscript to PLOS ONE. After careful consideration, we feel that it has merit but does not fully meet PLOS ONE’s publication criteria as it currently stands. Therefore, we invite you to submit a revised version of the manuscript that addresses the points raised during the review process.

We look forward to receiving your revised manuscript.

Kind regards,

Chen Zonghai

Academic Editor

PLOS ONE

Journal Requirements:

3. Our internal editors have looked over your manuscript and determined that it is within the scope of our Smart Energy Systems Call for Papers. The Collection will encompass the latest research in smart grid technologies, including information technologies, device integration, distribution methods, and data mining, all towards improving the efficiency of energy supply networks. Additional information can be found on our announcement page: https://collections.plos.org/call-for-papers/smart-energy-systems/. If you would like your manuscript to be considered for this collection, please let us know in your cover letter and we will ensure that your paper is treated as if you were responding to this call. If you would prefer to remove your manuscript from collection consideration, please specify this in the cover letter.

6. One of the noted authors is a group or consortium [ KalaignarKarunanidhi Institute of Technology Coimbatore – 641 402]. In addition to naming the author group, please list the individual authors and affiliations within this group in the acknowledgments section of your manuscript. Please also indicate clearly a lead author for this group along with a contact email address.

Additional Editor Comments:

Based on the comments of reviewers and the current level of the manuscript, it is recommended to make major modifications to the manuscript.

Reviewers' comments:

Reviewer's Responses to Questions

**Comments to the Author**

1. Is the manuscript technically sound, and do the data support the conclusions?

Reviewer #1: Yes

Reviewer #2: Partly

Reviewer #3: No

2. Has the statistical analysis been performed appropriately and rigorously? 

Reviewer #1: Yes

Reviewer #2: Yes

Reviewer #3: No

3. Have the authors made all data underlying the findings in their manuscript fully available?

Reviewer #1: No

Reviewer #2: No

Reviewer #3: No

4. Is the manuscript presented in an intelligible fashion and written in standard English?

Reviewer #1: Yes

Reviewer #2: No

Reviewer #3: No

5. Review Comments to the Author

Reviewer #1: This paper presents the Optimal Planning and Allocation of Plug-in Hybrid Electric Vehicles Charging Stations using a Novel Hybrid Optimization Technique. The paper can be accepted by addressing the following issues.

1. Please ensure all the variables have been defined correctly.

2. The reinforcement learning has been widely used for HEV and energy system optimization, e.g., DOI: 10.1109/TIE.2021.3070514; DOI: 10.1109/TII.2020.3014599. Such relevant works can be included for completeness and to enhance the literature review.

3. Please explain more about the validation condition.

4 The figures are not in good quality. Please modify.

5. The conclusion can be improved by giving the primary findings with necessary statistical results.

Reviewer #2: This paper offers an effective PHEV charging stations allocation approach for RES applications. The contributions and innovations of this paper are insufficient. Some concerns are listed below for possible further improvement of this paper.

1. The number of relevant literature introductions is relatively small, and half of them are articles published in 2018. It is suggested to add some new articles.

2. The introduction of relevant literature should be classified according to some logic, rather than simply spread out according to time.

3. How to ensure that the power grid can not only withstand the disorderly charging of charging piles but also ensure the charging efficiency?

4. How to calculate cost, VRPindex and Aindex in the objective function.

5. Please analyze the experimental results according to the principles of AT-AQ and other comparison algorithms, and explain why AT-AQ can achieve better experimental results.

6. Can you give the specific regional distribution of charging stations?

7. Noting that this study focuses on PHEV, is the proposed method applicable to EV charging in the future?

Reviewer #3: 1- Captions of most figures are inappropriate. Authors should give each figure an independent caption.

2- Grammar errors and typos should be avoided. Many words’ first letter is improperly capitalized.

3- The literature review is suggested to be merged to Section 1.

4- Most figures are of low quality.

5- As it is stated, “the maximum produced power of EM is limited to thirty kilowatts due to battery health.” Why? More detailed explanations should be given.

6- In Eqn. (5), objectives are with different units and magnitude orders. I do not think it is correct to merge them simply.

7- What are the constraints of the EV, AC/DC, and power plant in your system?

8- Comparative results in Section 5 should be fully discussed and analyzed.

9- EVs may leave the station after charging. However, EV arrival distribution during the experimental period is not introduced and considered.

10- In P15, “The analysis was carried out in two different scenarios.” However, in P17, three scenarios are utilized. Moreover, what are the specific differences among these scenarios?

6. PLOS authors have the option to publish the peer review history of their article (what does this mean?). If published, this will include your full peer review and any attached files.

Reviewer #1: No

Reviewer #2: No

Reviewer #3: No

---

## [Author Response · Author response to Decision Letter 0]

4 Mar 2023

RESPONSE TO REVIEWER COMMENTS

Manuscript No. - PONE-D-22-29934

Manuscript Title - Optimal Planning and Allocation of Plug-in Hybrid Electric Vehicles Charging Stations using a Novel Hybrid Optimization Technique

Authors - Ayyappan Subramaniam & Lal Raja Singh Ravi Sing

Reviewer #1: 

This paper presents the Optimal Planning and Allocation of Plug-in Hybrid Electric Vehicles Charging Stations using a Novel Hybrid Optimization Technique. The paper can be accepted by addressing the following issues.

1. Please ensure all the variables have been defined correctly.

In the revised manuscript, variables are defined as per the comments of the reviewer.

2. The reinforcement learning has been widely used for HEV and energy system optimization, e.g., DOI: 10.1109/TIE.2021.3070514; DOI: 10.1109/TII.2020.3014599. Such relevant works can be included for completeness and to enhance the literature review.

Based on the reviewer comments, the specified two literatures have been included in the literature review to enhance the review section. It is included in reference section at Sl.No. 34 & 35.

3. Please explain more about the validation condition.

Based on the reviewer comments, the validation condition has been detailed in the revised manuscript at the performance analysis section.

4 The figures are not in good quality. Please modify.

In respect of the reviewer comments, the quality of the Figures is improved in the revised manuscript submitted.

5. The conclusion can be improved by giving the primary findings with necessary statistical results.

Primary findings are included and the conclusion section is revised as per the reviewer comments.

Reviewer #2:  Some concerns are listed below for possible further improvement of this paper.

1. The number of relevant literature introductions is relatively small, and half of them are articles published in 2018. It is suggested to add some new articles.

With respect to the reviewer comments, new literatures pertaining to 2021 to 2023 has been included in detail in the Introduction section. These are also included in the references from Sl.no. 36 - 49. 

//The contents included based on reviewer comments are shown in RED colour in the revised manuscript//

2. The introduction of relevant literature should be classified according to some logic, rather than simply spread out according to time.

Yes, based on the reviewer comments, the literature section is remodified in section 1 of the revised manuscript.

3. How to ensure that the power grid can not only withstand the disorderly charging of charging piles but also ensure the charging efficiency?

This is because the voltage profile will be maintained substantially due to which the losses will be minimized increasing the charging efficiency. The power loss minimization enables to reduce the energy consumption and thereby the efficiency gets increased.

4. How to calculate cost, VRPindex and Aindex in the objective function.

Eq ([Disp-formula pone.0284421.e006]) in the revised manuscript, is employed to evaluate the necessary parameters.

5. Please analyze the experimental results according to the principles of AT-AQ and other comparison algorithms, and explain why AT-AQ can achieve better experimental results.

Based on the reviewer comments, the analysis is detailed in the revised manuscript at the performance analysis section.

6. Can you give the specific regional distribution of charging stations?

The dataset employed for testing and validating the proposed optimization model pertains to the usage of electric vehicles within the campus of Georgia Tech, Atlanta, USA and the vehicles were charged at the conference centre parking station and around 150 vehicles were flying around the campus. The average driving distance of the vehicles is 31 km. Thus the regional distribution will be around the campus of Georgia Tech, Atlanta, USA in this research study. 

7. Noting that this study focuses on PHEV, is the proposed method applicable to EV charging in the future?

Yes, the proposed approach is suitable for both PHEV and EV. The proposed optimization model is generic and shall be applied for both PHEV and EV due to their generalization ability and learning ability.

Reviewer #3: 

1- Captions of most figures are inappropriate. Authors should give each figure an independent caption.

Based on the reviewer comments, the captions of the figures are changed in the revised manuscript.

2- Grammar errors and typos should be avoided. Many words’ first letter is improperly capitalized.

Based on reviewer comments, English grammatical corrections are done.

3- The literature review is suggested to be merged to Section 1.

Considering the reviewer comments, the literature review section is merged with the Introduction section in the revised manuscript.

4- Most figures are of low quality.

In the revised manuscript, the quality of figures are improved.

5- As it is stated, “the maximum produced power of EM is limited to thirty kilowatts due to battery health.” Why? More detailed explanations should be given.

This is only for the reparative braking mechanism and the conditions of the battery is primarily important for this scenario. Due to which, for this braking mechanism only the maximum power produced is limited and for other forms of braking employed it can be extended to higher kilowatts.

6- In Eqn. (5), objectives are with different units and magnitude orders. I do not think it is correct to merge them simply.

Based on the reviewer comments, in the revised manuscript, Eq ([Disp-formula pone.0284421.e006]) provides the complete objective function employed in the proposed technique.

7- What are the constraints of the EV, AC/DC, and power plant in your system?

The constraints in respect of the EVs shall be AC power and necessary inverter circuits shall be employed for conversion of AC to DC and the DC power shall be stored in the batteries. Making the EVs into the power grid shall result in voltage drop, energy loss and affecting the peak load of the system. Thus, the constraints in respect of the EVs include,

-Cost of installation of the charging stations

-Increased distribution system power loss

-Difficulty in connecting the EVs for charging directly to the grid

-Problem in the source of electrical energy at a unity power factor and the voltage profile not maintained due to the power system module

-Increased power losses

-Active power loss of the distributed power system network

The above are the specific constraints pertaining to the effective location of charging stations for the Electric Vehicles. 

//The above contents are included in the section 3.1 of the revised manuscript based on the reviewer comments//

8- Comparative results in Section 5 should be fully discussed and analyzed.

Based on the reviewer comments, validation is included in the performance analysis section.

9- EVs may leave the station after charging. However, EV arrival distribution during the experimental period is not introduced and considered.

EV arrival distribution is always on the first come first serve basis and based on the free charging connectors. And this initialized in the algorithm as the node that arrives will be set to 1 and the charging station shall be allocated for the same. 

10- In P15, “The analysis was carried out in two different scenarios.” However, in P17, three scenarios are utilized. Moreover, what are the specific differences among these scenarios?

Three scenarios have been employed for this study, the scenarios include,

Scenario 1: Optimal placement based on distribution system conjunction with transportation system

Scenario 2: Optimal placement based on distribution generators with previous optimal charging load

Scenario 3: Allocation of distribution generators and charging station in distribution system optimally based on earlier optimal load 

The authors than the reviewers and the editor for their valuable comments in improving the quality of the manuscript. Thank you.

Sincerely,

Authors

---

## [Decision Letter · Decision Letter 1]

30 Mar 2023

Optimal Planning and Allocation of Plug-in Hybrid Electric Vehicles Charging Stations using a Novel Hybrid Optimization Technique 

PONE-D-22-29934R1

Dear Dr. Subramaniam,

We’re pleased to inform you that your manuscript has been judged scientifically suitable for publication and will be formally accepted for publication once it meets all outstanding technical requirements.

Kind regards,

Chen Zonghai

Academic Editor

PLOS ONE

Additional Editor Comments (optional):

Minor Revision

Reviewers' comments:

Reviewer's Responses to Questions

**Comments to the Author**

1. If the authors have adequately addressed your comments raised in a previous round of review and you feel that this manuscript is now acceptable for publication, you may indicate that here to bypass the “Comments to the Author” section, enter your conflict of interest statement in the “Confidential to Editor” section, and submit your "Accept" recommendation.

Reviewer #2: (No Response)

Reviewer #3: (No Response)

2. Is the manuscript technically sound, and do the data support the conclusions?

Reviewer #2: (No Response)

Reviewer #3: (No Response)

3. Has the statistical analysis been performed appropriately and rigorously? 

Reviewer #2: (No Response)

Reviewer #3: (No Response)

4. Have the authors made all data underlying the findings in their manuscript fully available?

Reviewer #2: (No Response)

Reviewer #3: (No Response)

5. Is the manuscript presented in an intelligible fashion and written in standard English?

Reviewer #2: (No Response)

Reviewer #3: (No Response)

6. Review Comments to the Author

Reviewer #2: The author has solved all the problems.The quality of the articles has improved. Recommended acceptance.

Reviewer #3: (No Response)

7. PLOS authors have the option to publish the peer review history of their article (what does this mean?). If published, this will include your full peer review and any attached files.

Reviewer #2: No

Reviewer #3: No

---

## [Editor Report · Acceptance letter]

4 Apr 2023

PONE-D-22-29934R1 

Optimal Planning and Allocation of Plug-in Hybrid Electric Vehicles Charging Stations using a Novel Hybrid Optimization Technique 

Dear Dr. Subramaniam:

I'm pleased to inform you that your manuscript has been deemed suitable for publication in PLOS ONE. Congratulations! Your manuscript is now with our production department. 

Kind regards, 

on behalf of

Prof. Chen Zonghai 

Academic Editor

PLOS ONE